# Pesticide-induced resurgence in brown planthoppers is mediated by action on a suite of genes that promote juvenile hormone biosynthesis and female fecundity

Yang Gao[1†], Shao-Cong Su[1†], Ji-Yang Xing[1†], Zhao-Yu Liu[1], Dick R Nässel[2], Chris Bass[3], Congfen Gao[1], Shun-Fan Wu[1,4]*

[1]State Key Laboratory of Agricultural and Forestry Biosecurity, College of Plant Protection, Nanjing Agricultural University, Nanjing, China; [2]Department of Zoology, Stockholm University, Stockholm, Sweden; [3]College of Life and Environmental Sciences, Biosciences, University of Exeter, Penryn, United Kingdom; [4]College of Sciences, Nanjing Agricultural University, Nanjing, China

*For correspondence:
wusf@njau.edu.cn

[†]These authors contributed equally to this work

Competing interest: The authors declare that no competing interests exist.

## eLife Assessment

This **useful** manuscript reports mechanisms behind the increase in fecundity in response to sublethal doses of pesticides in the crop pest, the brown plant hopper. The authors hypothesize that the pesticide works by inducing the JH titer, which through the JH signaling pathway induces egg development, for which the evidence was judged to be **solid**.

**Abstract** Pesticide-induced resurgence, increases in pest insect populations following pesticide application, is a serious threat to the sustainable control of many highly damaging crop pests. Resurgence can result from pesticide-enhanced pest reproduction; however, the molecular mechanisms mediating this process remain unresolved. Here we show that brown planthopper (BPH) resurgence in rice crops following exposure to sublethal doses of the pesticide emamectin benzoate (EB) results from the coordinated action of a suite of genes that regulate juvenile hormone (JH) levels, resulting in increased JH titer in adult females and enhanced fecundity. We demonstrate that EB treatment at sublethal levels results in profound changes in female BPH fitness, including increased egg maturation and oviposition. This enhanced reproductive fitness results from the EB-mediated upregulation of key genes involved in the regulation of JH, including *JHAMT and Kr-h1* and the downregulation of allatostatin (*AstA*) and allatostatin receptor (*AstAR*) expression. AstA signaling is known to inhibit the production of JH in the corpora allata and hence EB exposure diminishes this inhibitory action. We find that the changes in gene expression following EB exposure are caused by the allosteric action of this insecticide on its molecular target, the glutamate-gated chloride channel (GluClα). Collectively, these results provide mechanistic insights into the regulation of negative pesticide-induced responses in insects and reveal some key actors involved in the JH-signaling pathway that underpin pesticide resurgence.

## Introduction

Chemical pesticides remain the primary means of controlling many of the world's most damaging arthropod crop pests (*Janssen and van Rijn, 2021*; *Wu et al., 2018b*). However, pesticide applications can result in pest resurgence, increases in pest insect populations that exceed natural, untreated population sizes, following an initial reduction of the pest population (*Gandara et al., 2024*; *Guedes et al., 2016*; *Hardin et al., 1995*; *Wu et al., 2020*). Two mechanisms have been implicated in pest resurgence—the loss of beneficial insects including natural enemies and pesticide-enhanced reproduction in the pest insect (*Wu et al., 2020*). In the case of the latter, several pesticides, such as the insecticides triazophos, deltamethrin, and the fungicide jinggangmycin, have been reported to stimulate reproduction in pest insects (*Ge et al., 2010*; *Wu et al., 2018a*; *Zhang et al., 2014*). Pesticide-enhanced pest reproduction has been linked to changes in the physiology and biochemistry of pest organisms after exposure to pesticides (*Guedes et al., 2016*; *Wu et al., 2020*). However, the molecular mechanisms underlying enhanced reproduction associated with pest resurgence remain poorly resolved.

The brown planthopper (BPH), *Nilaparvata lugens* (Stål), is a notorious pest of rice crops throughout Asia causing annual losses of ~300 million dollars across major rice-producing countries (*Wu et al., 2020*; *Wu et al., 2018b*). BPH inhibits the growth of rice plants by feeding on the phloem and also transmits highly damaging plant viruses including rice grassy stunt virus and rice ragged stunt virus (*Sōgawa, 1982*). Currently, chemical insecticides play an indispensable role in the control of BPH due to their efficiency, rapid effect, and low cost. However, due to the widespread and intensive use of chemical insecticides, BPH has developed resistance to the majority of compounds used for control (*Wu et al., 2018b*; *Zeng et al., 2023*).

Emamectin benzoate (EB) and abamectin are avermectin pesticides and act as allosteric modulators of insect glutamate-gated chloride channels (GluClα) (*Wolstenholme, 2012*), thereby inhibiting muscle contractions that lead to the cessation of insect feeding and subsequent death (*Ishaaya et al., 2002*). These insecticides exhibit particularly strong effects against Lepidoptera such as the rice leaffolder, *Cnaphalocrocis medinalis* Guénee, an important foliage-feeding insect that attacks rice during the vegetative stage (*Chintalapati et al., 2016*). Both BPH and the rice leaffolder are migratory pests with overlapping migratory paths; however, their occurrence period in the field differs by approximately 1 month, with leaffolders appearing earlier than BPH. Therefore, the use of EB to control rice leaffolder has the potential to impact BPH arriving later via exposure to sublethal concentrations of this compound. It has been observed that when farmers use EB and abamectin to control leaffolders on rice crops in China, BPH outbreaks frequently occur in the same field. While sublethal doses of certain pesticides have been shown to enhance fecundity in BPH, including the insecticides triazophos and deltamethrin (*Ge et al., 2013*; *Ge et al., 2010*; *Zhang et al., 2014*; *Zhang et al., 2022b*) and the fungicides carbendazim and jinggangmycin (*Wu et al., 2018a*), whether avermectins trigger resurgence in BPH via insecticide-enhanced reproduction remains unclear.

Reproduction in insects is influenced by external factors such as light (*Wang et al., 2021*), temperature (*Meiselman et al., 2022*), humidity (*Roy et al., 2015*), and nutrition (*Smykal and Raikhel, 2015*), as well as endogenous factors such as juvenile hormone (JH) (*Santos et al., 2019*), ecdysone (*Hun et al., 2022*), insulin (*Ling and Raikhel, 2018*), and target of rapamycin (TOR) signaling pathways (*Ahmed et al., 2020*; *Badisco et al., 2013*; *Du et al., 2022*; *Lu et al., 2016*). Of these, JH has been particularly implicated in insecticide-induced enhanced fecundity, with triazophos and deltamethrin treatments leading to increased circulating JH III titers in BPH females (*Wu et al., 2020*). JH is synthesized and secreted by the corpora allata in insects (*Luo et al., 2021*), and promotes reproduction by regulating the biosynthesis and secretion of vitellogenin (Vg) in the female fat body, and stimulating the absorption of Vg by the developing oocytes (*Santos et al., 2019*). However, the regulation of JH is complex (*Riddiford, 2008*; *Santos et al., 2019*) and the key actors involved in JH-mediated pesticide-enhanced reproduction remain to be resolved. Previous findings reported that the GluCl receptor is involved in JH biosynthesis in cockroaches (*Chiang et al., 2002a*; *Chiang et al., 2002b*; *Liu et al., 2005*). However, it is still unknown whether insecticides that target the GluClα can increase reproduction in pest insects and the mechanisms by which GluClα receptors regulate JH production remain to be unraveled.

In this study, we used a range of approaches to investigate the impact of sublethal doses of avermectins on BPH fecundity and unravel the molecular mechanisms mediating enhanced reproduction

following exposure to this insecticide class. We show that avermectin exposure results in profound changes in the expression of a key suite of genes that in combination regulate JH, which results in increased JH titer in adult females, which promotes fecundity. Interestingly, we found that EB, known as an allosteric GluClα channel modulator, can downregulate AstA/AstAR expression and thereby promote JH production.

## Results

### Allosteric modulators (emamectin benzoate and abamectin) of the GluClα channel stimulate fecundity of female *N. lugens*

To investigate whether GluClα modulators affect fecundity in BPH, we first determined the sublethal and median lethal doses of EB on fourth-instar nymphs, as well as on newly emerged males and females of BPH (*Supplementary file 1*). For this, we employed two different bioassay methods, the rice seedling dip bioassay method and topical application bioassay method (*Wu et al., 2018b*; *Yao et al., 2017*), in order to assess both the systemic and contact toxicity of these insecticides (*Supplementary file 1*). We then systematically treated fourth-instar nymphs of BPH, newly emerged males and females with the estimated $LC_{15}$ or $LC_{50}$ concentrations of EB and examined the fecundity of BPH after these individuals mated with treated or untreated individuals. We use the term 't' to represent individuals treated with EB and 'ck' to indicate control individuals that were treated with insecticide diluent minus insecticide. First, we tested whether treatment of BPH with EB in the nymphal stage impairs the fecundity of BPH at the adult stage and thus exposed fourth-instar BPH nymphs to the pesticide. After exposure to the $LC_{15}$ and $LC_{50}$ concentrations of EB, the number of eggs laid per female of BPH in ♀t × ♂t crosses increased by 1.48- and 1.40-fold compared with control ♀ck × ♂ck crosses (*Figure 1A*); the number of eggs laid per female of BPH in ♀t × ♂ck crosses increased by 1.53- and 2.07-fold compared with control crosses (*Figure 1B*). However, the number of eggs laid per female of BPH in ♀ck × ♂t crosses did not increase significantly compared to control ♀ck × ♂ck crosses (*Figure 1C*).

Exposure of fourth-instar nymphs to the $LC_{15}$ and $LC_{50}$ concentrations of EB in contact bioassays also significantly stimulated fecundity. After treatment with the $LC_{15}$ and $LC_{50}$ concentrations of EB, the number of eggs laid per female of BPH in ♀t × ♂t crosses increased by 1.18- and 1.26-fold compared with the control (♀ck × ♂ck) (*Figure 1D*); the number of eggs laid per female of BPH in ♀t × ♂ck crosses increased by 1.27- and 1.56-fold compared with the control crosses (*Figure 1E*). However, there was no significant difference in the number of eggs laid between ♀ck × ♂t crosses and controls (♀ck × ♂ck) (*Figure 1F*). These results reveal that EB stimulates the fecundity of females following both systemic and contact routes of exposure.

Next, we examined whether EB treatment of adult BPH also enhances reproduction in relation to egg-laying. Indeed, treating newly emerged adults with the $LD_{15}$ and $LD_{50}$ concentrations of EB significantly stimulated the number of eggs laid per female (*Figure 1—figure supplement 1A*). Furthermore, sublethal exposure of fourth-instar BPH nymphs to another allosteric modulator of GluClα, abamectin ($LC_{15}$ and $LC_{50}$ concentrations), was also found to significantly enhance oviposition (*Figure 1—figure supplement 1B*).

To determine whether EB administered during the fourth-instar larval stage persists as residues in the adult stage, we used HPLC-MS/MS to quantify the amount of EB present at the adult stage after exposing fourth-instar nymphs to this compound. However, we found no detectable EB residues in the adult stage following fourth-instar nymphal treatment (*Figure 1—figure supplement 1C*). This suggests that the mechanism underlying the increased fecundity of female adults induced by EB treatment of nymphs may differ from that caused by direct EB treatment of female adults. Combined with our previous observation that EB treatment significantly increased the body weight of adult females (*Figure 1—figure supplement 3E and F*), a possible explanation for this phenomenon is that EB may enhance food intake in BPH, potentially leading to elevated production of insulin-like peptides and thus increased growth. Increased insulin signaling could potentially also stimulate JH biosynthesis during the adult stage (*Badisco et al., 2013*).

To examine if EB also stimulates egg-laying in other insect species, we conducted bioassays on the small BPH, *Laodelphax striatellus*, the white-backed planthopper, *Sogatella furcifera,* and vinegar flies, *Drosophila melanogaster*. In contrast to our findings in BPH, we found that sublethal doses ($LC_{15}$ and

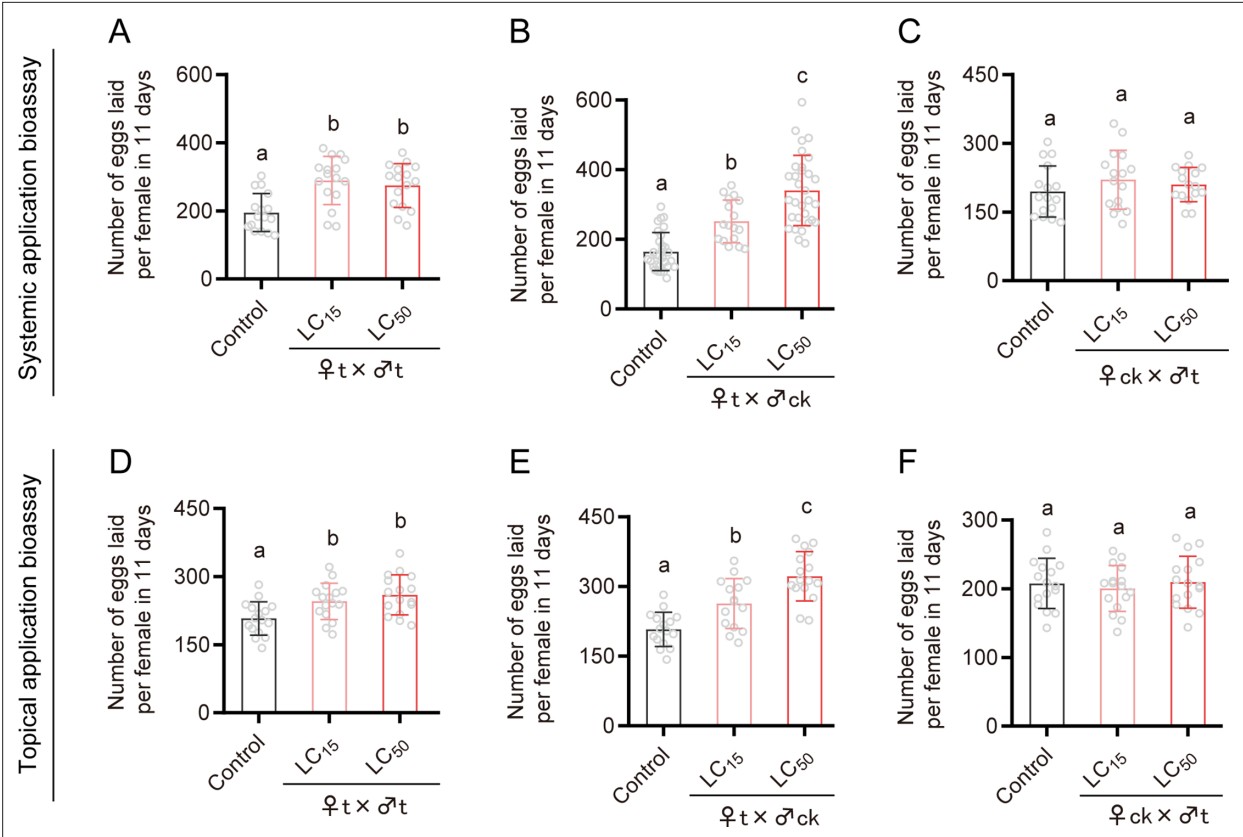

**Figure 1.** Fecundity of brown planthopper (BPH) following exposure to sublethal (LC$_{15}$) and median lethal (LC$_{50}$) concentrations of emamectin benzoate following systemic application bioassays (**A**: ♀t ×♂t; **B**: ♀t ×♂ck; **C**: ♀ck ×♂t) and topical application bioassays (**D**: ♀t ×♂t; **E**: ♀t ×♂ck; **F**: ♀ck ×♂t), respectively. The letter 't' represents treatment with insecticide, while 'ck' indicates controls that were not treated with insecticide. Insects were exposed as fourth-instar nymphs. (A) Left to right, n = 16, 16, and 16; (B) Left to right, n = 30, 17, and 31; (C) Left to right, n = 16, 16, and 16; (D) Left to right, n = 16, 16, and 16; (E) Left to right, n = 16, 15, and 16; (F) Left to right, n = 16, 16, and 16. All data are presented as the mean ± s.e.m. Different lowercase letters above the bars indicate significant differences (one-way ANOVA with Tukey's multiple range test, p<0.05).

The online version of this article includes the following source data and figure supplement(s) for figure 1:

**Source data 1.** Numerical source data for *Figure 1*.

**Figure supplement 1.** Effects of emamectin benzoate and abamectin on fecundity and residue dynamics in brown planthoppers.

**Figure supplement 1—source data 1.** Numerical source data for *Figure 1—figure supplements 1–3*.

**Figure supplement 2.** Fecundity of small brown planthopper, *Laodelphax striatellus* (**A–C**) white-backed planthopper, *Sogatella furcifera* (**D–F**) and vinegar fly, *Drosophila melanogaster* (**G, H**) when larvae and newly emerged adults were treated with sublethal concentrations of emamectin benzoate.

**Figure supplement 3.** The impact of emamectin benzoate on the reproductive fitness of brown planthopper (BPH).

LC$_{50}$) of EB inhibit fecundity of female *L. striatellus* (***Figure 1—figure supplement 2A–C***) and have no impact on the fecundity of *S. furcifera* (***Figure 1—figure supplement 2D–F***). In addition, we found that sublethal doses (LC$_{15}$, LC$_{30}$, or LC$_{50}$) of EB also inhibit fecundity in *D. melanogaster* (***Figure 1—figure supplement 2G and H***). These results indicate that the stimulation of reproduction by EB in BPH is species-specific and does not extend to even related insect species.

## The impact of EB treatment on BPH reproductive fitness

To better understand the effects of EB on the reproductive fitness of BPH, the duration of the preoviposition period, emergence rate, female ratio, female longevity, and female weight were evaluated following exposure using systemic bioassays. The preoviposition period of females treated with the LC$_{50}$ of EB decreased significantly compared with the control (***Figure 1—figure supplement 3A***). In contrast, no significant effects of EB on emergence rate and female ratio were observed (***Figure 1—figure supplement 3B and C***). Female survival analysis showed that exposure of fourth-instar nymphs

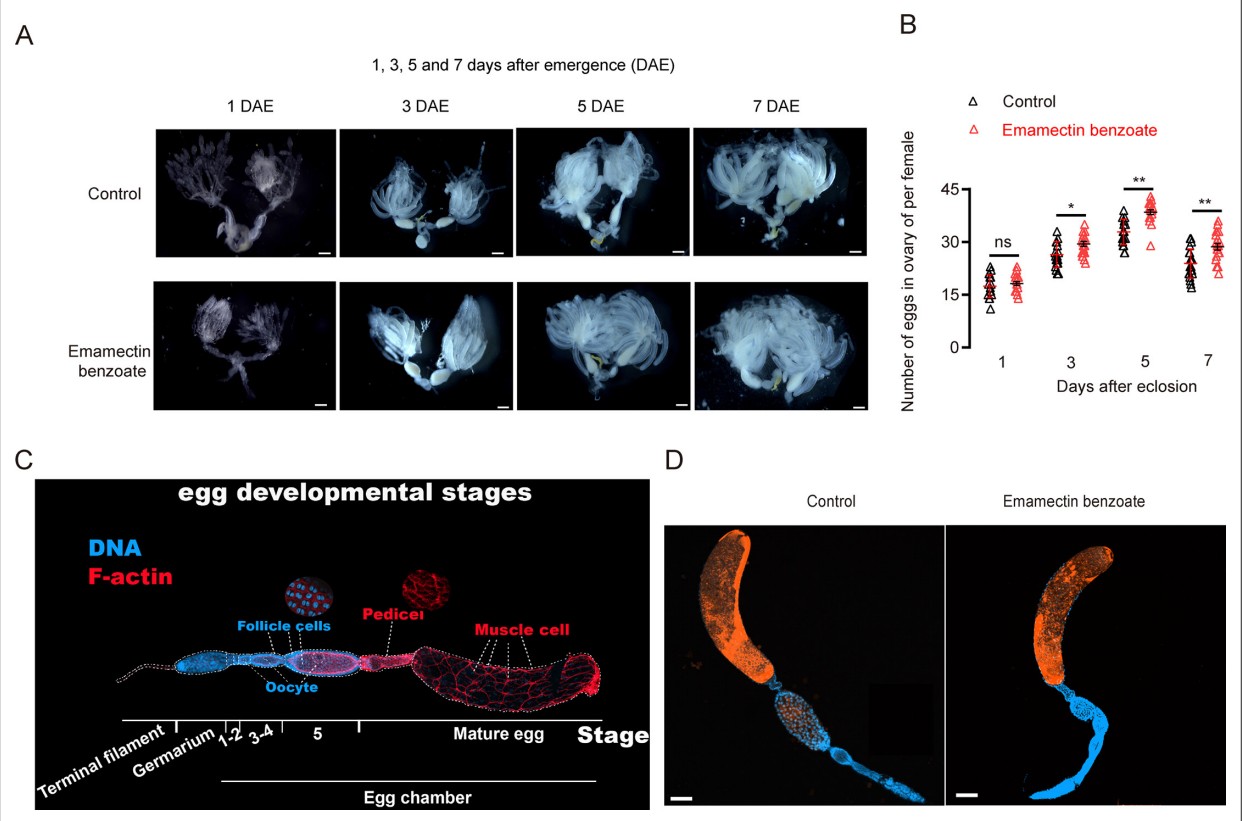

**Figure 2.** The impact of emamectin benzoate (EB) on ovarian maturation in brown planthopper (BPH). Fourth-instar nymphs were treated with the $LC_{50}$ concentration of EB in systemic bioassays. (**A**) Ovarian development in EB-treated BPH at 1, 3, 5, and 7 days after eclosion (DAE) compared to untreated controls. Scale bar = 1000 μm. (**B**) Number of mature eggs in the ovaries of EB-treated fourth-instar BPH nymphs measured at 1, 3, 5, and 7 DAE compared to controls. Left to right, n = 20, 19, 21, 21, 20, 20, 21, and 21. All data are presented as the mean ± s.e.m. Asterisks indicate values significantly different from the control using Student's *t*-test (ns, no significant; *p<0.05 and **p<0.01). (**C**) Different developmental stages of BPH eggs. (**D**) No impairment of EB on oogenesis of BPH. Scale bar = 100 μm.

The online version of this article includes the following source data and figure supplement(s) for figure 2:

**Source data 1.** Numerical source data for *Figure 2*.

**Figure supplement 1.** Number of mature eggs in the ovaries of emamectin benzoate (EB)-treated brown planthopper (BPH) female adults compared to controls.

**Figure supplement 1—source data 1.** Numerical source data for *Figure 2—figure supplements 1 and 3*.

**Figure supplement 2.** The oogenesis of brown planthopper (BPH).

**Figure supplement 3.** Amounts of glycogen (**A**), TAG (**B**), total protein content (**C**), cholesterol (**D**), and four circulating sugars including sucrose, glucose, fructose, and trehalose (**E–H**) after brown planthopper (BPH) exposure to emamectin benzoate (EB).

to the $LC_{50}$ of EB has no impact on female longevity (*Figure 1—figure supplement 3D*). Interestingly, the occurrence of brachypterism (short-wing phenotype) in females and female weight was significantly increased after EB exposure (*Figure 1—figure supplement 3E and F*). It was reported that short-winged females usually display higher fecundity than long-winged females in BPH, and two different insulin receptors determine the alternative wing morphs (*Xu et al., 2015*). However, whether insulin signaling is involved in the fecundity increase of BPH in our experiments requires further investigation (*Ling and Raikhel, 2021*; *Mirth et al., 2014*; *Sheng et al., 2011*).

## EB promotes ovarian maturation in BPH

To investigate the cause of increased egg-laying following EB exposure, we examined whether EB influences ovarian maturation in BPH. We dissected and compared the ovaries of females treated with the $LC_{50}$ of EB at 1, 3, 5, and 7 days after eclosion (DAE) with control females. At 3, 5, and 7 DAE, the number of eggs in the ovary of BPH in the EB-treated group was significantly higher than in controls

(*Figure 2A and B*). We furthermore observed that EB treatment of female adults also increases the number of mature eggs in the ovary (*Figure 2—figure supplement 1*). Hence, although EB does not affect the normal egg developmental stages (see description in next section), our results suggest that EB treatment promotes oogenesis and, as a result, the insects both produce more eggs in the ovary and a larger number of eggs are laid.

Unlike the model insect *D. melanogaster*, studies on oogenesis in BPH are relatively limited. Therefore, we used immunohistochemistry to stain eggs at different stages of BPH development. We captured images with a laser confocal microscope to document each stage of oogenesis in this species. We found that oogenesis in the BPH differs from that in *D. melanogaster* and belongs to the telotrophic meroistic type, where the oocyte is connected to nutrient-providing cells in the anterior trophic region of the ovariole through a nutritive cord. Oogenesis in the BPH occurs in three main compartments (*Figure 2—figure supplement 2*):

1. Terminal filament (TF): This region, composed of terminal filament cells, is similar to the terminal filament structure found in *Drosophila*.
2. Germarium: This region is where germline stem cells (GSCs) differentiate into oocytes and follicle cells. It typically consists of cap cells adjacent to the terminal filament, GSCs, escort cells, and cystoblasts.
3. Egg chamber: The egg chamber in the BPH primarily contains eggs at six developmental stages. The distinct layering of follicle cells, visible through cytoskeletal staining, is the main feature distinguishing the six different developmental stages listed in the legend of *Figure 2—figure supplement 2*.

We next explored whether EB treatment can enhance or impair oogenesis in BPH. However, dissection of various developmental stages revealed that EB treatment has no significant effects on developmental stages and morphology of oogenesis in BPH (*Figure 2C and D*). However, considering that the number of eggs laid by EB-treated females was larger than in control females (*Figure 1*, *Figure 1—figure supplement 1*), our data indicates that EB treatment of BPH can both promote oogenesis and oviposition.

## EB exposure increases circulating sugars and the storage of macromolecules in BPH

Nutritional status is an important indicator of reproductive fitness. Thus, to investigate whether EB affects intermediary metabolism and energy storage in BPH, glycogen, triacylglyceride (TAG) total protein content, cholesterol, and four circulating carbohydrates were quantified in fourth-instar BPH nymphs following exposure to the $LC_{50}$ of EB.

We found that EB exposure has no impact on glycogen levels (*Figure 2—figure supplement 2A*). The amount of TAG in EB-treated BPH was 27% higher (p<0.05) than those in controls, but only in BPH of the late fifth instar (5L) stage, with no significant differences observed in subsequent developmental stages (*Figure 2—figure supplement 2B*). The amount of total protein content in EB-treated BPH was higher than the control groups in the case of all developmental stages from 5L nymph to 7 DAE (*Figure 2—figure supplement 2C*). EB exposure also increased cholesterol levels at 4 and 5 DAE (*Figure 2—figure supplement 2D*). Compared with the solvent control, EB treatment caused significant increases (p<0.05) in the levels of sucrose, glucose, fructose, and trehalose (*Figure 2—figure supplement 2E–H*). Thus, collectively, these data provide evidence that EB exposure leads to energy mobilization such as an increase in circulating lipids and carbohydrates in BPH.

## EB stimulates egg-laying that is mediated by the JH signaling pathway

Given the important role of JH in vitellogenesis and egg development in insects (*Jing et al., 2021*; *Ling and Raikhel, 2021*; *Luo et al., 2021*; *Santos et al., 2019*; *Zheng et al., 2022*), we asked whether EB treatment influences the titer of JH in BPH. As measured by ELISA, the JH titer of BPH nymphs treated with the $LC_{50}$ concentration of EB was significantly lower than that of controls in systemic bioassays during the middle and late stages of the fourth instar (*Figure 3A*). However, at 2, 3, and 4 DAE, the JH titer in the EB treated group was significantly higher than that of the control (*Figure 3A*). Interestingly, the titer of another important insect hormone, the steroid 20-hydroxyecdysone, was not significantly different between EB-treated BPH and solvent-treated controls (*Figure 3—figure supplement 1*). To independently validate the results of the ELISA, we employed HPLC-MS/MS to

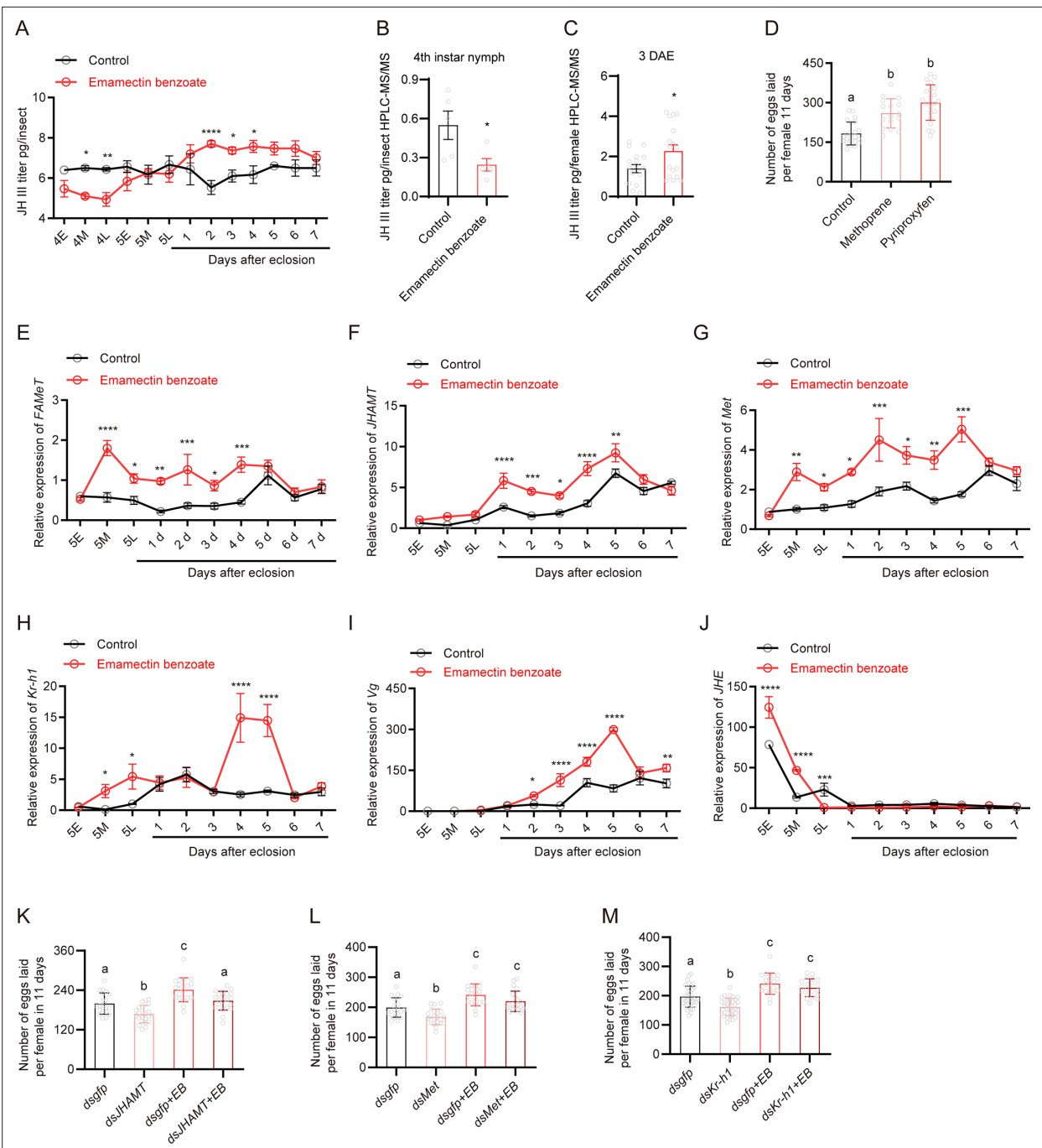

**Figure 3.** Emamectin benzoate (EB)-induced reproduction in brown planthopper (BPH) is mediated by components of the juvenile hormone (JH) signaling pathway. (**A**) The titer of JH III (as measured by ELISA) at different developmental stages of whole-body BPH when fourth-instar nymphs were treated with the median lethal concentrations of EB. N = 3 for each point. (**B, C**) The titer of JH III (as measured by HPLC-MS/MS) in whole-body BPH females at 4L and 3 days after eclosion (DAE) when treated with median lethal concentrations of EB. N = 5 for panel B and n = 17 for panel C. (**D**) Oviposition rate of BPH when fourth-instar nymphs were treated with 4 ppm methoprene or 10 ppm pyriproxyfen. Left to right, n = 22, 14 and 21. (**E–J**) Expression of selected JH-related genes (*FAMeT*, *JHAMT*, *Met*, *Kr-h1*, *Vg*, and *JHE*) in EB-treated BPH. N = 8 for panel E-I and n = 10 for panel J. (**K**) Egg production following silencing of *JHAMT* with or without EB application. Left to right, n = 7, 7, 7 and 7. (**L**) Egg production following silencing of *met* with or without EB application. Left to right, n = 7, 8, 7 and 7. (**M**) Egg production after silencing *Kr-h1* with or without EB application. Left to right, n = 7, 7, 7 and 7.All data are presented as means ± s.e.m. Student's *t*-test was used to compare controls and treatments. One-way ANOVA with Tukey's multiple comparisons test was used to compare more than two samples. ns, no significant difference; asterisks indicate values significantly different from the control (ns, no significant; *p<0.05, **p<0.01, ***p<0.001, and ****p<0.0001). Different lowercase letters above the bars indicate significant differences (p<0.05).

*Figure 3 continued on next page*

*Figure 3 continued*

The online version of this article includes the following source data and figure supplement(s) for figure 3:

**Source data 1.** Numerical source data for *Figure 3*.

**Figure supplement 1.** 20-Hydroxyecdysone titer at different developmental stages of whole-body brown planthopper (BPH) when fourth-instar nymphs were treated with median lethal concentrations of emamectin benzoate (EB).

**Figure supplement 1—source data 1.** Numerical source data for *Figure 3—figure supplements 1–4*.

**Figure supplement 2.** Effects of *Kr-h1* RNAi knockdown on *Kr-h1* and *Vg* expression in brown Planthopper (BPH).

**Figure supplement 3.** Effects of emamectin benzoate on hormone titers and gene expression in brown planthopper female adults.

**Figure supplement 4.** Evaluation of the potential of emamectin benzoate (EB) to reverse dsRNA-mediated silencing by quantifying Kr-h1 gene expression.

measure the JH titer in BPH following EB exposure (*Bownes and Rembold, 1987*; *Guo et al., 2020*; *Jing et al., 2021*). The results showed that the JH III titer significantly decreased after EB treatment at the late fourth-instar nymph stage (*Figure 3A and B*), but significantly increased at the third day after eclosion (3 DAE) (*Figure 3A and C*). We also investigated the effects of EB treatment on the JH titer of female adults. The data indicate that the JH titer was also significantly increased in the EB-treated female adults compared with controls (*Figure 3—figure supplement 3*). However, again the steroid 20-hydroxyecdysone was not significantly different between EB-treated BPH and controls (*Figure 3—figure supplement 3*).

To further investigate the role of JH in EB-enhanced fecundity in BPH, we treated BPH with methoprene and pyriproxyfen, JH analogs, or biologically active JH mimic, respectively, to determine whether they can stimulate fecundity. Both compounds significantly increased egg-laying in BPH (*Figure 3D*). Taken together, these results reveal that EB exposure is associated with an increase in JH titer and that this elevated JH signaling contributes to enhanced fecundity in BPH.

Since we found that EB could induce JH synthesis in the BPH, we asked whether EB could influence the expression of genes that are involved in JH biosynthesis or degradation. For this, we treated fourth-instar nymphs or newly emerged female adults of BPH with the $LC_{50}$ concentration of EB using systemic bioassay, and then collected early (5E), middle (5M), and late (5L) stage of fifth-instar nymph and 1–7 DAE female adults for analysis. Quantitative PCR was then used to examine the expression of key genes previously implicated in the regulation of JH (*Gospocic et al., 2021*; *Nouzova et al., 2018*).

Farnesoic acid O-methyltransferase (FAMeT) (*Liu et al., 2010b*) is known as an important enzyme in the JH biosynthetic pathway, catalyzing methylation of farnesoic acid (FA) to methyl farnesoate (MF) (*Nouzova et al., 2018*). We found that this gene was significantly upregulated in 5M instar nymphs to 4 DAE following EB treatment (1.0-fold to 3.0-fold) (*Figure 3E*). Juvenile hormone acid methyltransferase (JHAMT) (*Nouzova et al., 2018*; *Shinoda and Itoyama, 2003*), which is involved in the biosynthesis of JH, was also upregulated in EB-treated BPH at 1–5 DAE compared to controls (1.5-fold to 3.0-fold) (*Figure 3F*). We furthermore found that EB treatment in female adults increases JHAMT expression (*Figure 3—figure supplement 3*).

Methoprene-tolerant (Met) belongs to the basic helix–loop–helix Per/Arnt/Sim (bHLH-PAS) family of transcription factors and is the intracellular (nuclear) receptor of JH (*Riddiford, 2020*; *Zhu et al., 2019*). The levels of *met* mRNA slightly increased in EB-treated BPH at the 5M and 5L instar nymph and 1–5 DAE adult stages compared to controls (1.7-fold to 2.9-fold) (*Figure 3G*). However, it should be mentioned that JH action does not result in an increase of *Met*. Thus, it is possible that other factors (indirect effects) induced by EB treatment cause the increase in the mRNA expression level of *Met*.

Krüppel homolog 1 (*Kr-h1*), a transcriptional target of JH signaling, is reported to be sensitive to levels of JH, and its expression levels are directly correlated with JH titers (*Gospocic et al., 2021*; *Jindra et al., 2013*; *Zhang et al., 2022a*). We found that *Kr-h1* was significantly upregulated in the adults of EB-treated BPH at the 5M, 5L nymph and 4–5 DAE stages (4.7-fold to 27.2-fold) when fourth-instar nymph or female adults were treated with EB (*Figure 3H*, *Figure 3—figure supplement 3D*).

Similarly, the expression level of *vitellogenin* (*Vg*), a key downstream component of JH signaling that triggers ovarian development in insects, including BPH (*Shen et al., 2019*), was markedly increased in females at 2–5 DAE by EB (1.7-fold to 5.5-fold) (*Figure 3I*). During the maturation process, eggs require uptake of vg, and an increase in vg content can accelerate egg maturation, producing more

mature eggs. Since our molecular data suggest that EB treatment leads to an upregulation of *vg* expression, we propose that the increase in egg-laying caused by EB treatment is at least in part due to the upregulation of *vg* (*Figure 3I*), which elevates Vg content, leading to increased uptake of Vg by maturing eggs and resulting in the production of more mature eggs.

Juvenile hormone esterase (JHE) is the primary JH-specific degradation enzyme that plays a key role in regulating JH titers (*Kamita et al., 2003*). Interestingly, we observed a significant upregulation of JHE mRNA levels in the early and middle fifth-instar nymph stage, followed by downregulation in 5L instar nymphs of EB-treated BPH (1.3-fold to 2.6-fold) (*Figure 3J*).

Taken together, these results suggest that EB has a profound impact on the expression of key genes involved in the biosynthesis of JH or genes in the downstream signaling pathway genes that might promote egg development and female fecundity. However, our data do not allow us to conclude whether the effect of EB on the genes regulating the JH titer is direct or indirect.

To further understand whether these JH pathway-related genes are involved in egg-laying behavior in BPH, we performed RNAi experiments to downregulate the expression of *Kr-h1* (*Figure 3—figure supplement 2A*). We found that silencing of *Kr-h1* diminished *vg* gene expression (*Figure 3—figure supplement 2B*). Importantly, RNAi-induced diminishment of *JHAMT*, *Met*, and *Kr-h1* genes in female BPH was also found to suppress egg-laying (*Figure 3K–M*). However, this phenotype was rescued by EB treatment, which restored egg-laying to normal levels in BPH injected with *JHAMT*, *Met*, and *Kr-h1* dsRNA (*Figure 3K–M*). Note that this rescue is possible since knockdown of the genes is incomplete when using dsRNA injection (and residual gene expression allows for EB action). Together these results provide a mechanistic understanding of how EB enhances fecundity in BPH by modulating the expression of key genes involved in JH synthesis.

Next, we investigated whether EB treatment could rescue the dsRNA-mediated gene silencing effect. To address this, we selected the *Kr-h1* gene and analyzed its expression levels after EB treatment. Our results showed that *Kr-h1* expression was suppressed by ~70% at 72 h post-dsRNA injection. However, EB treatment did not significantly rescue *Kr-h1* expression in gene knockdown insects (*$p$* > 0.05 compared with *dsKr-h1* treatment) at 24 h post-EB treatment (*Figure 3—figure supplement 4*). While dsRNA-mediated *Kr-h1* suppression was robust initially, its efficacy may decline during prolonged experiments. This aligns with reports in BPH, where effects of RNAi gradually diminish beyond 7 days post-injection (*Liu et al., 2010a*). The late-phase fecundity increase might reflect partial *Kr-h1* recovery due to RNAi degradation, allowing residual EB to weakly stimulate reproduction. In addition, the physiological impact of EB (e.g., neurotoxicity, hormonal modulation) could manifest via compensatory feedback loops or metabolic remodeling.

## EB induces JH biosynthesis through the peptidergic AstA/AstAR signaling pathway

In many insect species, the timing and level of JH biosynthesis can be precisely regulated by neuropeptides, such as stimulatory allatotropins (ATs) and inhibitory allatostatins (Asts), (*Bellés et al., 1999*; *Kataoka et al., 1989*; *Kramer et al., 1991*; *Lorenz et al., 1995*; *Stay and Tobe, 2007*; *Verlinden et al., 2015*; *Wegener and Chen, 2022*; *Woodhead et al., 1989*). Insects can, in a species-specific manner, produce one type of AT and three types of Asts: FGL-amide Ast (AstA) (*Woodhead et al., 1989*; *Yin et al., 2006*); myoinhibitory peptide (MIP or AstB) (*Stay and Tobe, 2007*) and PISCF Ast (AstC) (*Wang et al., 2012*). In some species, there also exist two paralogous genes of AstCs that are named *AstCC* and *AstCCC* (*Veenstra, 2009*; *Veenstra, 2016*). Interestingly, the allatostatic activity of these three types of Ast peptides varies between insect species so that in each species only one type of Ast (e.g., AstA) controls JH production (*Nässel, 2002*; *Wang et al., 2012*; *Stay and Tobe, 2007*; *Wegener and Chen, 2022*).

Analysis of neural transcriptome sequence data of BPH has revealed the presence of one AT, four types of Asts, and four corresponding receptors, allatotropin receptor (A16, ATR), AstA receptor (A2, AstAR), AstB (MIP) receptor (A10, AstBR, or MIPR), and AstC receptor (A1, AstCR) (*Tanaka et al., 2014*). We cloned the five neuropeptide genes (encoding AT, AstA, AstB/MIP, AstCC, and AstCCC) and confirmed the sequences obtained from transcriptome data (*Figure 4—figure supplement 1*; *Tanaka et al., 2014*). Interestingly, we found that AstC is missing in the genome of BPH and only AstCC and AstCCC are present (*Figure 4—figure supplement 1*). Next, we also cloned their corresponding receptors (*Veenstra, 2009*), including ATR (A16), AstAR (A2), AstBR (A10), and AstCR (A1),

which might be activated by AstCC and/or AstCCC (*Audsley et al., 2013*; *Veenstra, 2009*; *Zhang et al., 2022a*). Sequence alignments and phylogenetic analysis are shown in *Figure 4—figure supplement 2*.

Quantitative PCR was then used to examine whether EB treatment influences the expression of the genes encoding these neuropeptides and their receptors. Treating fourth-instar or female adult BPH with the $LC_{50}$ concentration of EB significantly increased the expression of *AT*, *ATR*, and *AstCCC*, while resulting in the downregulation of *AstA*, *AstB/mip*, *AstCC*, AstAR, and *AstBR/mipr* at the adult stage (*Figure 4A and B*, *Figure 4—figure supplement 3*). Among these, *AstA* and *AstAR* were the most downregulated genes after EB treatment (*Figure 4A and B*, *Figure 4—figure supplement 3H*) and thus the *AstA/AstAR* signaling system was selected for subsequent functional analysis. Silencing of *AstAR* in female BPH using RNAi (*Figure 4C*) resulted in an increased number of eggs laid per female compared with *dsgfp*-injected controls (*Figure 4D*). Interestingly, silencing *AstAR* also resulted in the upregulation of *JHAMT*, *Met,* and *Kr-h1,* which are involved in the JH biosynthesis/signaling (*Figure 4E–H*). However, *JHE* was not influenced by AstAR silencing (*Figure 4I*). We next investigated whether silencing the *AstAR* gene influences the JH titer in BPH. Indeed, a significantly increased JH titer was observed in *AstAR*-silenced BPH compared with controls (*Figure 4J*). Thus, our data strongly suggest that AstA is a key inhibitor of JH production in BPH.

Finally, we investigated whether injection of mature Ast and AT peptides could influence the number of eggs laid and the JH titer in BPH. We synthesized one AT, six AstAs (AstA1 to AstA6), one AstCC, and one AstCCC peptide according to their determined sequences (*Figure 4—figure supplement 1*). Indeed, we found that injection of AstA1 and AstA6 in female adults reduced the number of eggs laid per female over 48 h (*Figure 4K*). We also observed that AstA1 injection decreased the JH titer as measured 16 h and 48 h after injection (*Figure 4L*). AT injection increased the JH titer after 2 h, but levels returned to normal 4 h after injection (*Figure 4M*). Collectively, our data provide compelling evidence that EB enhances reproduction in BPH probably indirectly through the AstA/AstAR and JH signaling pathways and further supports the role of AstA and AT in regulation of the JH titer in this species.

## EB-enhanced fecundity in BPH is dependent on its molecular target protein, the GluClα channel

EB and abamectin are allosteric modulators, which target glutamate-gated chloride channels (GluCls) (*Sparks et al., 2020*; *Sparks et al., 2021*; *Wolstenholme, 2012*; *Wu et al., 2017*). This channel is found in invertebrates only and appears to be monomeric and formed by GluClα subunits (*Wolstenholme, 2012*). Hence, we examined whether EB-stimulated fecundity in BPH is mediated by its molecular target Gluclα. The full-length *Gluclα* coding sequence from BPH was cloned and sequenced (*Figure 5—figure supplement 1*), and the impact of EB on *Gluclα* gene expression was examined using quantitative PCR. Treatment of BPH with the $LC_{50}$ concentration of EB significantly downregulated *Gluclα* gene expression at the 5E and 5M nymph stages, while upregulating *Gluclα* gene expression at 2 DAE and 5 DAE in the adult stage (*Figure 5A*). We also found that EB treatment in female adults also increases the *Gluclα* gene expression (*Figure 5—figure supplement 3*). However, we do not propose that EB is a direct transcriptional regulator of *Gluclα* since EB and other avermectins are known to alter the channel conformation and thus their function (*Wolstenholme, 2012*; *Wu et al., 2017*). Thus, it is likely that the observed increase in *Gluclα* transcript is a secondary effect downstream of EB signaling.

To examine the role of *Gluclα* gene in BPH fecundity, RNAi was used to knock down expression of this gene in female adult BPH (*Figure 5B*). A significant decrease in the number of eggs laid per female was observed in *dsGluClα*-injected insects compared with *dsgfp*-injected insects (*Figure 5C*). However, treatment with EB was found to partially rescue the decreased egg-laying phenotype induced by *dsGluClα* injection (*Figure 5C*). Note that this rescue is possible since knockdown of the genes is incomplete when using dsRNA injection (and residual gene expression allows for EB action). To investigate the mechanism by which *Gluclα* expression affects fecundity, we examined if silencing *Gluclα* influences JH titer and JH-associated gene expression. Indeed, we observed that RNAi knockdown of *Gluclα* leads to a decrease in JH titer (*Figure 5D*) and downregulation of genes including *JHAMT* which is responsible for JH synthesis, and the JH signaling downstream genes *Met* and *Kr-h1* (*Figure 5E–G*). In contrast, expression of *FAMeT* and *JHE* was not changed in the *Gluclα* knockdown

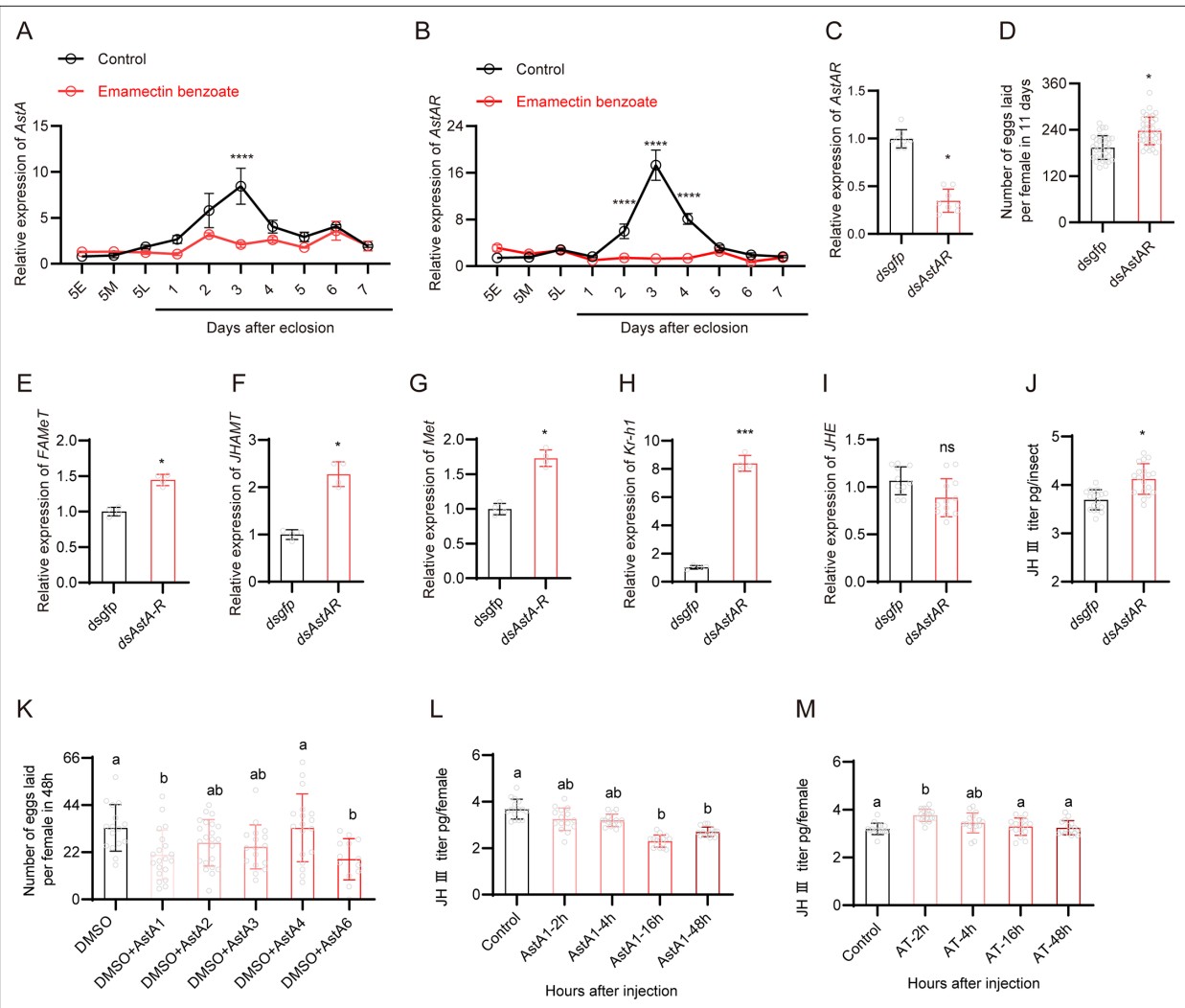

**Figure 4.** Emamectin benzoate (EB)-induced reproduction in brown planthopper (BPH) is mediated by the AstA/AstAR and juvenile hormone (JH) signaling pathway. (**A, B**) Expression of *AstA* and *AstAR* in different stages of BPH following EB treatment. N = 8 for panel A and n = 12 for panel B. (**C**) Downregulation of *AstAR* using RNAi leads to a reduction in mRNA expression level. Left to right, n = 8 and 8. (**D**) Egg production in female BPH following silencing of *AstAR* gene. Left to right, n = 47 and 49. (**E–I**) Expression of selected JH signaling pathway related genes (*JHAMT*, *Met*, *Kr-h1*, and *JHE*) in *AstAR* silenced BPH. N = 4 for panel E-H. Left to right, n = 11 and 12 for panel I. (**J**) JH III titer of BPH females after *AstAR* gene silencing determined by HPLC-MS/MS. Left to right, n = 18 and 18. (**K**) Number of eggs laid per female in 48 h following injection of the six mature AstA1-AstA6 peptides and one mature AT peptide. Fifty nanoliters of PBS (as control) and seven different peptides (20 pmol/insect) were injected into female BPH 3 days after eclosion. Left to right, n = 16, 20, 22, 17, 18, 13, and 14. (**L, M**) The JH III titer of whole-body BPH females at different time points following AstA or AT injection. N = 15 for panel L and n = 16 for panel M. All data are presented as means ± s.e.m. Student's *t*-test was used to compare controls and treatments. One-way ANOVA with Tukey's multiple comparisons test was used to compare more than two samples. ns, no significant difference; asterisks indicate values significantly different from the control (ns, no significant; *p<0.05, **p<0.01, ***p<0.001, and ****p<0.0001). Different lowercase letters above the bars indicate significant differences (p<0.05).

The online version of this article includes the following source data and figure supplement(s) for figure 4:

**Source data 1.** Numerical source data for *Figure 4*.

**Figure supplement 1.** Alignments of the amino acid sequences of (**A**) AT, (**B**) AstA, (**C**) AstB/MIP, (**D**) AstCC, and (**E**) AstCCC peptides from select species.

**Figure supplement 2.** Phylogenetic tree of the predicted brown planthopper (BPH) (*) allatotropin receptor (A16, ATR), allatostatins A receptor (A2, AstAR), AstB (MIP) receptor (A10, AstBR or MIPR), and allatostatins C receptor (A1, AstCR) with other insect species.

**Figure supplement 3.** Emamectin benzoate (EB)-induced changes in the expression of *AT* (**A**, n = 8), *AstB* (**B**, n = 4), *AstCC* (**C**, n = 4), *AstCCC* (**D**, n = 4), *ATR* (**E**, n = 4), *AstBR* (**F**, n = 4), and *AstCR* (**G** and **H**, n = 4) in brown planthopper (BPH).

**Figure supplement 3—source data 1.** Numerical source data for *Figure 4—figure supplement 3*.

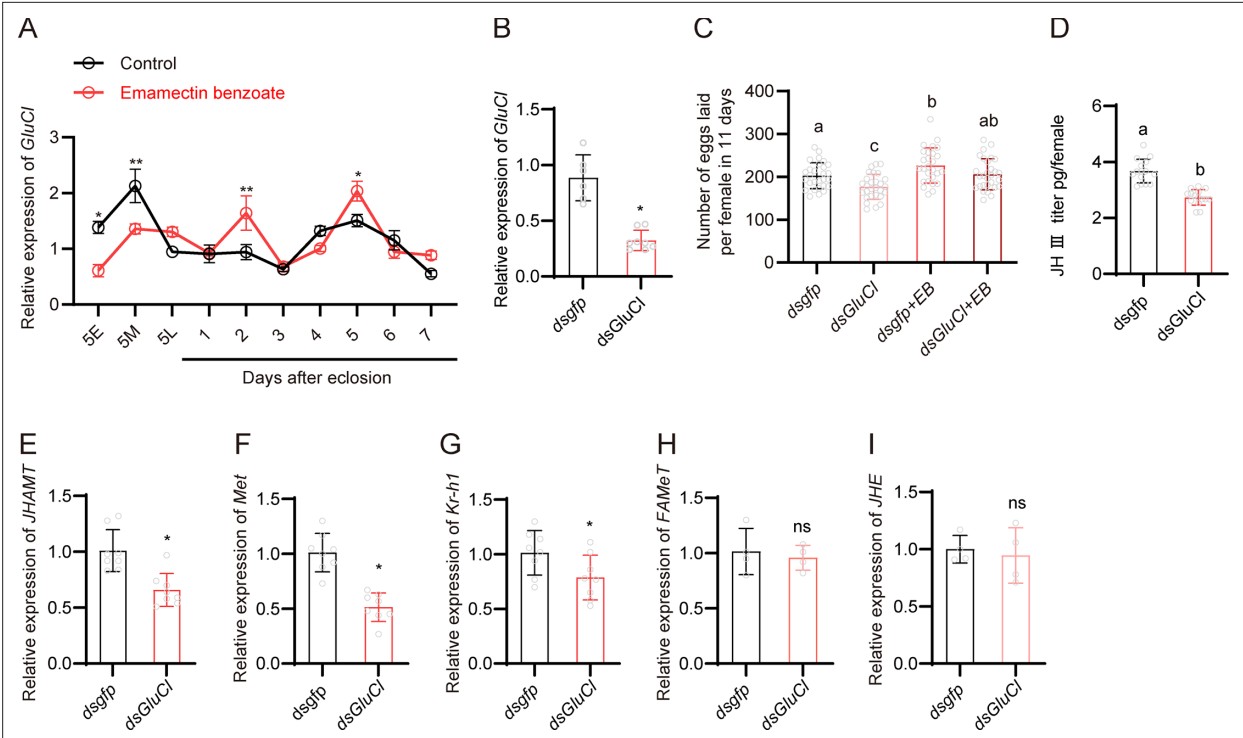

**Figure 5.** Role of emamectin benzoate (EB) and the GluCl channel in fecundity and juvenile hormone signaling in brown planthopper (BPH).
(**A**) Expression of *GluCl* in EB-treated and untreated BPH. N = 8 for each point. (**B**) Expression of *GluCl* following injection of *dsGluCl* in BPH. Left to right, n = 6 and 8. (**C**) Egg production after *GluCl* gene knockdown in EB-treated and untreated BPH. Left to right, n = 28, 28, 27, and 29. (**D**) The juvenile hormone (JH) III titer of whole-body BPH females after *GluCl* gene silencing as quantified using the ELISA method. Left to right, n = 15 and 15. (**E–I**) Expression patterns of selected JH-related genes (*JHAMT, Met, Kr-h1, FAMeT and JHE*) in *GluCl* silenced BPH. N = 8 for panel E–G and n = 4 for panel H and I. All data are presented as means ± s.e.m. Student's *t*-test was used to compare the two samples. One-way ANOVA with Tukey's multiple comparisons test was used to compare more than two samples. ns, no significant difference; asterisks indicate values significantly different from the control (ns, no significant; *$p<0.05$ and **$p<0.01$). Different lowercase letters above the bars indicate significant differences ($p<0.05$).

The online version of this article includes the following source data and figure supplement(s) for figure 5:

**Source data 1.** Numerical source data for *Figure 5*.

**Figure supplement 1.** Phylogenetic analysis of glutamate-gated chloride channels in different species.

**Figure supplement 2.** The expression of *AT* (**A**, n = 8), *AstA* (**B**, n = 8), *AstB* (**C**, n = 4), *AstCC* (**D**, n = 4), *AstCCC* (**E**, n = 4), *AstAR* (**F**, n = 8), *AstBR* (**G**, n = 4), and *AstCR* (**H**, n = 4) in BPH injected with *dsGluCl or dsgfp*.

**Figure supplement 2—source data 1.** Numerical source data for *Figure 5—figure supplements 2 and 3*.

**Figure supplement 3.** The expression of *GluCl* after female adult brown planthopper (BPH) was treated with emamectin benzoate (EB).

insects (*Figure 5H and I*). We also examined whether silencing *Gluclα* impacts the *AstA/AstAR* signaling pathway in female adults. Knockdown of *Gluclα* in female adults was found to have no impact on the expression of *AT, AstA, AstB, AstCC, AstAR*, and *AstBR*. However, the expression of *AstCCC* and *AstCR* was significantly upregulated in *dsGluclα*-injected insects (*Figure 5—figure supplement 2A–H*). Further studies are required to delineate the direct or indirect mechanisms underlying this effect of *Gluclα*-knockdown.

## Discussion

Pesticide-induced resurgence of pest insects is a serious problem in rice and several other crops (*Wu et al., 2020*). However, the mechanisms underpinning pesticide-enhanced reproduction in insects remain poorly understood. Here we reveal that a suite of molecular actors underlie this trait that, in combination, mediate profound physiological changes in the reproductive fitness of female BPH. Our data provide fundamental insights into the molecular mechanisms by which xenobiotics modify insect

reproduction and have implications for devising strategies for the control of a highly damaging crop pest. We discuss these topics below.

## Sublethal doses of the GluCl modulators, EB and abamectin, stimulate fecundity in BPH

We show that in both contact and systemic assays the avermectins EB and abamectin stimulate reproduction in BPH. Thus, insecticide-enhanced reproduction is likely a key factor in the BPH resurgence observed when farmers use EB and abamectin to control leaffolders in rice crops in China (*Yao et al., 2017*). Although this is the first report of sublethal doses of avermectins enhancing insect fecundity, our findings are consistent with previous studies, which have shown that certain other insecticides, herbicides, and fungicides stimulate BPH reproduction (*Cheng et al., 2014*; *Ge et al., 2011*; *Ge et al., 2013*; *Wang et al., 2010*; *Wu et al., 2020*; *Yang et al., 2019*; *Wan et al., 2013*; *Zhang et al., 2014*; *Zhao et al., 2011*). Intriguingly, we show that EB only induces fecundity in female adults and is specific to BPH, with EB exposure failing to enhance reproduction in two related species, the small BPH, *L. striatellus* and the white-backed planthopper, *S. furcifera*, or the model insect *D. melanogaster*. Thus, the mechanisms underpinning this trait appear to act exclusively on female BPH and may be specific to this species. Nevertheless, our findings are of general interest since they link the molecular EB target, the GluCl channel, and JH signaling to fecundity. Furthermore, our preliminary experiments suggest that EB diminishes fecundity in *Drosophila* (see *Figure 1—figure supplement 2G and H*), a finding worth pursuing in this genetically tractable insect. It can be noted that the GluCl channels were first identified and cloned in the nematode worm *Caenorhabditis elegans*, where they were also found to be targets of avermectins and involved in reproduction and fecundity (*Wolstenholme, 2012*).

Pesticides may stimulate insect reproduction through a variety of physiological and molecular mechanisms. Our data reveal that exposure to sub-lethal concentrations of EB results in profound changes to female BPH fitness, leading to increases in female weight, contents of total protein, cholesterol, and sugars, as well as egg production and decreases in duration of the preoviposition period. Some of these findings exhibit parallels with previous studies, which demonstrated that treating third-instar BPH nymphs with either deltamethrin, triazophos, or imidacloprid led to increased soluble sugar levels in the corresponding adults (*Yin et al., 2008*). Such metabolites provide the energy that drives BPH reproduction and hence resurgence (*Wu et al., 2020*). Thus, together with prior work, our results suggest that pesticides associated with resurgence stimulate nutritional intake in BPH to fuel enhanced energy-intensive reproduction.

## The JH signaling pathway plays an essential role in EB-induced fecundity in BPH

JH is a pleiotropic hormone, which plays important roles in development and reproduction in insects (*Jindra et al., 2013*; *Riddiford, 2008*; *Santos et al., 2019*). Circulating JH titers are regulated by factors that control JH production in the corpora allata including biosynthetic enzymes and catabolic enzymes that regulate JH levels. Our results show that EB exposure is associated with increased circulating JH III titers in BPH females over 2–4 DAE and promotes ovarian development. Previous studies have reported that triazophos and deltamethrin treatments also lead to increased circulating JH III titers in BPH females over 1–3 days post emergence. Similarly, jinggangmycin treatments were found to lead to increased JH titers (by approximately 45–50%) in BPH females over 2 days post emergence (*Xu et al., 2016*). Thus, our findings, in combination with these previous studies, demonstrate that insecticide treatments can have dramatic effects on the regulation of key insect hormones involved in pest reproduction, which can in turn drive pesticide resurgence.

Although increased JH titers following pesticide exposure have been correlated with reduced levels of active JH esterase during the first 3 days PE (*Ge et al., 2010*), the types and numbers of mechanisms mediating the observed increase in hormone titer have remained an open question. Our data reveal that the elevated JH titer in EB-exposed BPH is associated with the upregulation of a gene that encodes a biosynthetic enzyme for JH (*JHAMT*) and a downstream signaling gene that can induce *vg* gene expression (*kr-h1*). Using RNAi, we provide experimental evidence of the role of these genes in the regulation of JH III and fecundity of female BPH and demonstrate that EB can restore the reduction in egg production resulting from the partial RNAi knockdown of *JHAMT*, *met,* and *kr-h1* expression.

JHMAT is an enzyme that catalyzes the conversion of inactive precursors of JH to active JH in the final stages of JH biosynthesis (*Nouzova et al., 2018*; *Shinoda and Itoyama, 2003*). Interestingly, while it has not been previously implicated in pesticide resurgence, treatment of the stored product pest *Sitotroga cerealella* with diallyl trisulfide, an insecticidal compound in garlic essential oil, was found to increase JHAMT mRNA levels (*Shah et al., 2022*). Because JHMAT is the key rate-limiting enzyme in the regulation of JH titer, our results suggest that its enhanced expression is a key molecular mechanism of pesticide resurgence in BPH.

Met is a ligand-activated member of the basic helix–loop–helix Per/Arnt/Sim (bHLH-PAS) transcription factors and is the intracellular receptor for JH (*Charles et al., 2011*; *Jindra et al., 2015*; *Riddiford, 2020*; *Zhu et al., 2019*). Kr-h1 is a zinc finger protein that acts downstream of Met and is expressed in response to JH signaling. Although the genes encoding these proteins have not been previously linked to pesticide resurgence, our finding that they are upregulated following EB exposure and demonstration of their role in promoting fecundity is consistent with previous studies (*Santos et al., 2019*). Specifically, treatment of BPH with JH III or the insecticidal analogs methoprene or pyriproxyfen was found to induce the expression of *Kr-h1* (*Jin et al., 2014*). Furthermore, knockdown of *Met* and *Kr-h1* in brachypterous BPH females was found to result in delayed ovariole maturation, and this was significantly more pronounced than the response observed in BPH treated separately with ds*NlMet* or ds*NlKr-h1* (*Lin et al., 2015*). This finding provides evidence of a possible interaction between *Met* and *Kr-h1* and, in combination with our data, suggests that *Met* and *Kr-h1* may act in concert to mediate EB-enhanced fecundity. However, our results showed that EB treatment can weakly increase (about twofold) expression of the *Met* gene in BPH (*Figure 3G*). Our data furthermore indicates that *Met* and *FAMeT* expression levels were not drastically influenced by EB compared with *kr-h1* and *vg* (*Figure 3H and I*). It should be mentioned that JH action does not directly result in the increase of *Met*. However, we cannot rule out the possibility that other factors (indirect effects), induced by EB treatment, increase the mRNA expression level of *Met*. One recent paper reported that downregulation of transcription factor *CncC* increases met expression in beetles (see *Figure 6A* in the reference; *Jiang et al., 2023*). Furthermore, several studies have reported that insecticide treatment activates the *CncC* gene signaling pathway, which regulates detoxification gene expression (*Amezian et al., 2023*; *Fu et al., 2024*; *Hu et al., 2021*). Hence, it is possible that EB might influence the *CncC* gene pathway, which in turn induces *met* expression.

Notably, after exposing fourth-instar BPH nymphs to EB, no EB residues were detected in the subsequent adult stage. This finding indicates that the EB-induced increase in adult fecundity is initiated during the nymphal stage and manifests in adulthood—a mechanism distinct from the direct fecundity enhancement of fecundity observed when EB is applied to adults. We propose that sublethal EB exposure during critical nymphal stages may reprogram metabolic or endocrine pathways, potentially via insulin/JH crosstalk. For instance, increased nutrient storage (e.g., proteins, sugars; *Figure 2—figure supplement 2*) could enhance insulin signaling, which in turn promotes JH biosynthesis in adults (*Ling and Raikhel, 2021*; *Mirth et al., 2014*; *Sheng et al., 2011*). Future studies should test whether EB alters insulin-like peptide expression or signaling during development.

## The EB-induced increase in fecundity in BPH is dependent on the allatostatin A signaling pathway

In addition to regulatory proteins that promote JH production, insects utilize peptides that regulate JH biosynthesis. These include the inhibitory allatostatins: FGLamides (FGLa; AstA), the W(X)6Wamides (AstB), and the PISCFs (AstC) (*Nässel, 2002*; *Stay et al., 1996*; *Stay and Tobe, 2007*; *Tanaka et al., 2014*; *Wegener and Chen, 2022*; *Zhang et al., 2021*). Interestingly, we found that EB exposure results in a marked downregulation of the expression of the genes encoding the allatostatin *AstA* and its receptor *AstAR*. We provide evidence for the functional impact of this on JH synthesis and BPH fecundity by (i) demonstrating that RNAi knockdown of *AstAR* expression results in increased JH titer and enhanced egg production, and (ii) showing that injection of female BPH with synthetic AstA peptide reduces the JH titer and decreases egg production. Thus, our data provide unequivocal evidence that AstA is a key inhibitor of JH production in BPH. This finding is consistent with previous work that has shown that FGLa/ASTs (AstA) inhibit JH biosynthesis in cockroaches and termites (*Woodhead et al., 1989*; *Yagi et al., 2005*). To our knowledge, our study is the first report of insecticides

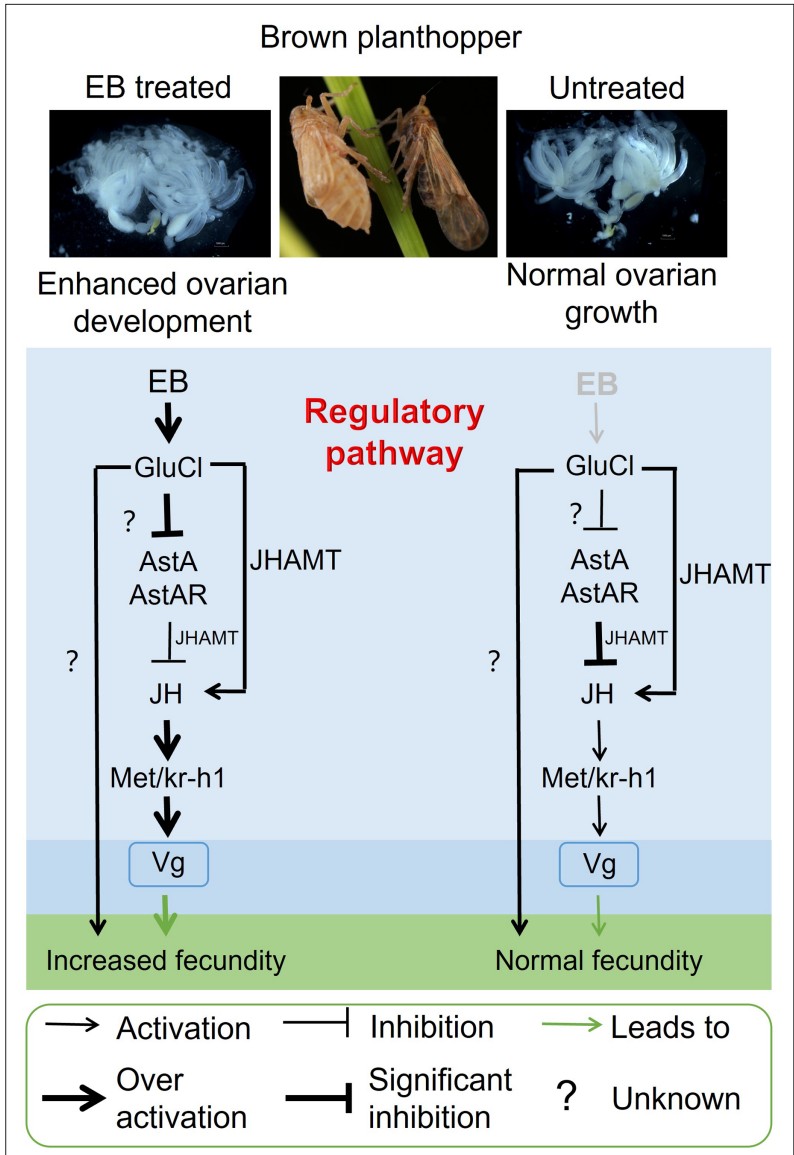

**Figure 6.** Schematic of the proposed regulatory pathway of emamectin benzoate (EB)-enhanced fecundity in brown planthopper (BPH). EB exposure results in the upregulation of genes that promote juvenile hormone (JH) signaling pathway (*JHAMT and Kr-h1*) and the downregulation of genes that inhibit it (allatostatin, Ast*A* and allatostatin A receptor, *AstAR*). This transcriptome reprogramming is dependent on the allosteric action of EB on its molecular target the glutamate-gated chloride channel (GluCl) receptor. Note that the mechanism of action on the GluCl is unknown in BPH, but likely the channel conformation changes and renders the receptor dysfunctional. Importantly, we do not suggest that the GluCl upregulation is due to direct action of EB on the channel. The resulting increased JH titer promotes vitellogenin (vg) biosynthesis and increased fecundity in EB-exposed insects. We observe significant cross-talk in the expression of genes that inhibit JH production and those that promote it, with *AstAR* inhibiting the expression of *JHAMT, Met,* and *Kr-h1* and *GluCl* activating the expression of *JHAMT*, which is responsible for JH synthesis, and the JH signaling downstream genes *Met and Kr-h1*.

downregulating the expression of the neuropeptide receptor, *AstAR*, and linking this to increases in JH titer and enhanced reproduction in insects.

Interestingly knockdown of the *AstAR* resulted in significant increases in the expression of genes involved in JH biosynthesis/signaling, including *JHAMT, Met,* and *Kr-h1*. Related to this finding, previous work has shown that knockdown of the AstA receptor gene, *Dar-2*, in *D. melanogaster* results in changes in the expression of genes encoding *Drosophila* insulin-like peptides (DILPs) and adipokinetic hormone (AKH) (*Hentze et al., 2015*). Together with our findings, this demonstrates that AstA

receptors may modulate the expression of numerous downstream genes involved in metabolism, energy storage, and reproduction. In the case of pesticide resurgence, our findings suggest that EB may act directly and/or indirectly on several genes that affect JH signaling and fecundity.

## The GluCl plays a role in EB-induced fecundity in BPH

EB and abamectin are allosteric modulators of GluCls (*Sparks et al., 2020*; *Sparks et al., 2021*; *Wu et al., 2017*). Our data revealed that EB exposure increases the expression of the GluCl in the adult stage of BPH, and knockdown of GluCl expression resulted in a reduction in both JH levels and egg production and egg-laying. This EB action on *GluClα* expression is likely indirect, and we do not consider EB as a transcriptional regulator of *GluClα*. Thus, the mechanism behind EB-mediated induction of *GluClα* remains to be determined. It is possible that prolonged EB exposure triggers feedback mechanisms (e.g., cellular stress responses) to counteract EB-induced *GluClα* dysfunction, leading to transcriptional upregulation of the channel. Hence, considering that EB exposure in our experiments lasts several days, these findings might represent indirect (or secondary) effects caused by other factors downstream of GluCl signaling that affect channel expression.

There are several studies showing that insects treated with insecticides display increases in the expression of target genes. For example, the relative expression level of the ryanodine receptor gene of the rice stem borer, *Chilo suppressalis,* was increased 10-fold after treatment with chlorantraniliprole, an insecticide that targets the ryanodine receptor (*Peng et al., 2017*). In *Drosophila*, starvation (and low insulin) elevates the transcription level of the receptors of the neuropeptides short neuropeptide F and tachykinin (*Ko et al., 2015*; *Root et al., 2011*). In BPH, reduction in mRNA and protein expression of a nicotinic acetylcholine receptor α8 subunit is associated with resistance to imidacloprid (*Zhang et al., 2015*). Knockdown of the α8 gene by RNA interference decreased the sensitivity of *N. lugens* to imidacloprid (*Zhang et al., 2015*). Hence, the expression of receptor genes may be regulated by diverse factors, including insecticide exposure.

Interestingly, the GluCl has been reported to inhibit the biosynthesis of JH in the cockroach, *Diploptera punctata* (*Liu et al., 2005*). Recent work has also shown that modulation of glutamatergic signals may contribute to the photoperiodic control of reproduction in bean bug, *Riptortus pedestris* (*Hasebe and Shiga, 2022*). Furthermore, work on *D. punctata* has revealed that application of the GluCl channel agonist ivermectin, which like EB belongs to the avermectin family, caused a decline in JH synthesis in the corpus allatum (*Liu et al., 2005*). While the inhibitory effect of ivermectin observed in this previous study differs from the activating effect of EB we observed in our study, it implicates a role for the GluCl channel in the regulation of JH regulation.

## Conclusion

Our study has revealed that exposure of BPH to EB affects a diverse suite of genes that act in combination to enhance the JH titer and thus increase female fecundity. A schematic of how these factors promote ovarian maturation in the adult stage of *N. lugens* through the AstA and JH signaling pathway is provided in . Our findings provide the foundation for further work to understand how these genes interact and the mechanisms by which their expression is activated or repressed by EB. Furthermore, our findings provide fundamental insights into the molecular response in insects to xenobiotic stress and illustrate that pesticides can have unexpected and positive impacts on pest populations. In this regard, our findings also have applied implications for the control of a highly damaging crop pest. Previous studies have reported that avermectins such as abamectin are toxic to the wolf spider *Pardosa pseudoannulata*, which is the main predator of BPH in rice crops (*Chen et al., 2017*; *Sogawa, 2015*). Thus, these insecticides both stimulate reproduction in BPH and kill their natural enemies, providing a 'perfect storm' for inducing damaging BPH outbreaks. Based on these findings, to avoid BPH resurgence, we suggest that the agents, EB and abamectin, should not be (or rarely be) applied to rice plants at growth stages when BPH are present. On a more optimistic note, our findings have identified numerous genes that play key roles in BPH reproduction and thus represent promising targets for the development of novel controls against this important pest.

# Materials and methods

## Insects

BPH was initially collected from Wanan, Jiangxi Province, in 2020, reared on 'Taichung Native 1' (TN1) rice seedlings in the laboratory without exposure to any insecticides. The strain was maintained in a climatic chamber at 27 ± 1°C, with relative humidity of 70 ± 10% and a light:dark = 16 h:8 h photoperiod.

## Chemicals

EB (95.2%) was obtained from Hebei Weiyuan Co., Ltd. (Hebei, China). Abamectin (96.8%) was obtained from Hebei Weiyuan Co., Ltd. (Hebei, China). Pyriproxyfen (98%) was obtained from Shanghai Shengnong Co., Ltd. (Shanghai, China). Methoprene (S)-(+) (mx7457-100mg) was purchased from Shanghai MaoKang Biotechnology Co., Ltd., (Shanghai, China). JH standard sample (J912305-10mg) was purchased from Shanghai Macklin Biotechnology Co., Ltd., (Shanghai, China).

## Bioassay

Different life stages of insects were used to perform bioassays to investigate the effects of insecticide on nymphs and adults. To test whether treatment of the nymph stage of insects would promote reproduction in females, we used fourth-instar nymphs of BPH or third-instar nymphs of *L. striatellus* and *S. furcifera* to perform bioassays. To test whether treatment of the adult stage of insects would promote reproduction in females, we used newly emerged male and female BPH.

### Systemic bioassays

The rice-seeding dipping bioassay method was used to evaluate the susceptibility of BPH, *L. striatellus,* and *S. furcifera* to EB. Technical-grade insecticides were dissolved in acetone as stock solution, then diluted in a series of six concentrations with water containing 0.1% Triton. Selected rice seedlings at the 6–8 cm growth stage were dipped in insecticide solutions for 30 s and then air-dried at room temperature. The roots of the rice seedlings were wrapped with cotton strips, and seedlings were placed in a plastic cup 5 cm in diameter. Fifteen insects were introduced into each plastic cup for each replicate. The top of the cup was then sealed with gauze to prevent escape. All experiments comprised at least three biological replicates. Control rice seedings were treated with 0.1% Triton X-100 water solution only. All treatments were maintained under standard conditions of 27 ± 1°C and 70–80% relative humidity with a 16 h light/8 h dark photoperiod. Mortality was assessed after 4 days for *N. lugens* or 2 days for *L. striatellus* and *S. furcifera* after treatment with insecticides. The insects were considered dead if they were unable to move in response to gentle prodding with a fine brush.

For *Drosophila* larval bioassays, we adopted a method described previously in our lab with minor modifications (*Huang 黄镜梅 et al., 2020*). Briefly, 23rd-instar larvae were placed in fly vials containing fly food (based on corn powder, brown sugar, yeast, and agar) supplemented with EB of different concentrations. Four concentrations ($LC_{10}$, $LC_{30}$, and $LC_{50}$) were tested together with a negative (no insecticide) control. For *Drosophila* adult bioassays, we selected virgin females 3 DAE. Several concentrations were overlaid onto fly food in standard *Drosophila* vials and allowed to dry overnight at room temperature. A total of 15 adult flies (3 DAE) were then added to each vial, and mortality was assessed after 2 days. Four replicates were carried out for each concentration. Control mortality was assessed using vials containing food with solvent minus insecticide.

### Contact bioassays

For topical bioassays, working insecticide solutions were prepared in acetone. Fourth-instar nymphs or newly emerged males/females were anesthetized with carbon dioxide for 5 s, and then 0.04 µl/insect test solution applied topically to the dorsal plates with a capillary micro-applicator (*Yao et al., 2017*). Insects were then placed in an artificial climate incubator with a temperature of 27 ± 1°C, a photoperiod of 16:8 h (L:D), and a humidity of 70% ± 10%. Mortality was determined 2 days after treatment. Data with over 20% mortality in the control treatment were discarded, and the assay was repeated at least three times.

## Fecundity assays

Fourth-instar nymphs of BPH and third-instar nymphs of *L. striatellus* and *S. furcifera* were treated with the $LC_{15}$ and $LC_{50}$ of EB, and then transferred to fresh rice seedlings. After eclosion, the adults were

used in the following experiment. Newly emerged treated adults and untreated adults were paired to produce four groups: untreated males and untreated females (♂ck×♀ck; ck indicates untreated); untreated males and treated females (♂ck×♀t; t indicates insecticide treated); treated males and untreated females (♂t×♀ck); and treated males and treated females (♂t×♀t). Each group comprised at least 10 mating pairs. All pairs were transferred to glass cylinders (diameter 5 cm and height 20 cm) containing rice plants (25 days old almost 20 cm high) as a food source for 11 days. The number of eggs and nymphs in plants was counted under a microscope.

For *Drosophila* egg-laying assays, we adopted our previous method (*Wu et al., 2019*). Briefly, insecticide-treated virgin females were paired with untreated males for 3 days, and then the mated females were transferred into the *Drosophila* ovipositing apparatus. The number of eggs was counted after 16 h.

## Fitness analysis

The fitness of EB-treated BPH was analyzed using methods reported previously (*Zeng et al., 2023*). We selected two groups, (♂ck×♀ck) and (♂ck×♀t), to study the effects of the LC$_{50}$ concentration of EB on BPH fitness. In the case of systemic exposure, fourth-instar nymphs of BPH were treated with the LC$_{50}$ of EB for 4 days and then transferred to tubes containing untreated rice plants for individual rearing. The rice plants were replaced every 3 days with untreated plants. The emergence ratio, female ratio, preoviposition period, female longevity, brachypterism female ratio and female weight were calculated.

## Examination of ovarian development

Adult females from ♂ck×♀ck control and ♂ck×♀t group on 1, 3, 5, and 7 DAE were dissected to observe ovarian development. The mature eggs in the ovary were photographed and recorded. Each group comprised at least 15 replicates.

To examine whether EB treatment impairs egg maturation, we dissected untreated or EB-treated ovaries and fixed them in 4% paraformaldehyde in phosphate-buffered saline (PBS) for 30 min at room temperature. After four washes of 10 min (4×10 min) in PAT3 (PBS with 0.5% Triton X-100 and 0.5% bovine serum albumin), the ovaries were then incubated with DAPI (4',6-diamidino-2-phenylindole, 100 nM) and Actin-stain 670 Fluorescent Phalloidin (200 nM). Imaging was performed using a Zeiss LSM980 confocal laser microscope.

## Measurements of glycogen, triglyceride, total protein content, cholesterol, and four sugars

The content of glycogen, triglyceride, cholesterol, and total protein was determined by spectrophotometry at 620 nm, 510 nm, 510 nm, and 562 nm, respectively, using the glycogen assay kit (A043-1-1), triglyceride reagent kit (A110-1-1), cholesterol assay kit (A111-1-1), and total protein assay kit (A045-2-2) obtained from the Nanjing Jiancheng Bioengineering Institute following the manufacturer's instructions. The determined results were normalized to the protein content in the sample, which was determined using BCA Protein Assay Reagent Kit (Thermo Scientific, Waltham, USA). Each sample contained tissue extracts from five adult female BPH, with three biological replicates per sample.

To assess the content of four sugars (sucrose, glucose, fructose, and trehalose) in the extract of BPH tissue, the same extraction method was used as above. Sugar content was quantified using the colorimetric method by scanning at 290 nm, 505 nm, 285 nm, and 620 nm, respectively, using the sucrose reagent kit (A099-1-1), glucose reagent kit (F006-1-1), fructose reagent kit (A085-1-1), and trehalose reagent kit (A150-1-1) obtained from the Nanjing Jiancheng Bioengineering Institute based on the manufacturer's instructions. Each sample contained tissue extracts from five adult female *N. lugens*, with three biological replicates per sample.

## Determination of JH III and 20-hydroxyecdysone titers of BPH by ELISA

The titer of JH III in BPH was measured using the JH ELISA Kit (Lot Number: HLE92086, Shanghai HaLing Biotechnology Co., Ltd., Shanghai, China) that employs the dual-antibody sandwich ELISA method. The titer of 20-hydroxyecdysone in BPH was measured using the 20-hydroxyecdysone ELISA

Kit (Lot Number: ZK14705, Shanghai ZhenKe Biotechnology Co., Ltd., Shanghai, China). At least three biological replicates were employed for each treatment.

## Determination of JH III titer in BPH using HPLC-MS/MS

The whole bodies of five individual BPH were mixed with 1 ml of n-hexane, followed by centrifugation at 10,000×$g$ for 10 min, and the upper hexane layer was then dried with nitrogen, dissolved in methanol, and sonicated for 10 min; after centrifugation at 10,000×$g$ for 10 min, the supernatant was collected through the organic filter membrane of 0.22 μm into 2 ml vials for JH III determination. JH III standard sample (J912305-10 mg) purchased from (Shanghai McLean Biochemical Technology Co. Ltd), dissolved in methyl alcohol as stock solution 10,000 mg/L was diluted in a series of six concentration gradients to serve as a reference. Liquid chromatography-tandem mass spectrometry (LC-MS/MS) was then carried out using UPLC Xevo TQ-S Micro (Waters Technology), quantitative method according to the external standard, the chromatographic column was EC-C18 (4.6 mm×150 mm, 2.7 μm), column temperature was 30°C, injection volume was 20 μl, elution flow rate was 0.3 ml/min, the mobile phase was acetonitrile:formic acid water (90:10), detection wavelength was 218 nm, the peak height was used for quantification.

## Determination of EB content using HPLC-MS/MS

4L BPH were systemically treated with EB as outlined above, samples were collected at 3 days post-treatment and transferred to healthy rice seedlings for 3 more days, after which another sample was collected. Each experiment included three biological replicates, with approximately 20 BPHs weighed per replicate. The weight of each sample was recorded, and 400 μl of methanol was added to each sample. The samples were then ground for 180 s at 65 Hz, followed by ultrasonic treatment at 4°C for 30 min. Next, 400 μl of n-hexane was added, and the mixture was vortexed to mix thoroughly. The samples were centrifuged at 12,000 rpm for 15 min at 4°C, and the supernatant was collected into a centrifuge tube and evaporated to dryness in a concentration evaporator. Then, 200 μl of methanol was added for reconstitution, followed by vortex mixing. The samples were centrifuged again at 12,000 rpm for 15 min at 4°C, and the supernatant was analyzed using the LC-MS/MS system. The EB standard was purchased from Shanghai Minxin Biotechnolog. Six concentration gradients were used for elution. The column temperature was maintained at 40°C, and the flow rate was set to 0.300 ml/min. LC-MS/MS was performed using a Waters Acquity UPLC system and an AB SCIEX 5500 QQQ-MS mass spectrometer. The chromatographic column used was Acquity UPLC BEH C18 (1.7 μm, 2.1 mm × 100 mm). A standard curve was generated for content calculation.

## Cloning, sequence, and phylogenetic analysis

The NCBI database and BLAST program were used to carry out sequence alignment and analysis. Open-reading frames (ORFs) were predicted with EditSeq. Primers were designed using the primer design tool in NCBI. Total RNA extraction was extracted from 30 adults BPH using TRIzol reagent (Invitrogen, Carlsbad, CA, USA) according to the manufacturer's instructions. cDNA was synthesized using the Biotech M-MLV reverse transcription kit. Full-length gene sequences were amplified by PCR using cDNA as template and Rapid Taq Master Mix (Vazyme Biotech, Cat# P222-02). The PCR product was purified on a 1% agarose gel, cloned into pClone007 Simple Vector Kit (Tsingke Biotech, Cat# TSV-007S), and then sequenced using the 3730 XL DNA analyzer (Applied Biosystems, Carlsbad, CA, USA). *Supplementary file 2* contains a list of the primers used in this study.

The exon and intron architectures of *AT*, *AstA*, *AstB*, *AstCC*, and *AstACCC* were predicted based on the alignments of putative ORFs against their corresponding genomic sequences. Sequence similarity/annotations and orthologous gene searches were performed using BLAST programs available in NCBI. Multiple alignments of the complete amino acid sequences were performed with Clustal Omega (http://www.ebi.ac.uk/Tools/msa/clustalo). Phylogeny was conducted using the maximum likelihood technique to create phylogenetic trees, and these were bootstrapped with 1000 replications using the MEGA 6 software (*Tamura et al., 2013*).

## RNA interference

Double-stranded RNA (dsRNA) of *gfp* (green fluorescent protein), *JHAMT* (JH acid O-methyltransferase), *Met* (methoprene-tolerant), *Kr-h1* (krüppel homolog 1), *AstAR* (allatostatin-A receptor), and

*GluCl* (glutamate-gated chloride channel) was prepared using Ambion's MEGAscript T7 kit following the manufacturer's instructions. The primer sequences for double-stranded RNA synthesis are listed in *Supplementary file 2*. Newly emerged females were injected with 40 nl (5000 ng/µl) of double-stranded RNA of *gfp* (*dsgfp*) or double-stranded RNA of the target genes in the conjunctive part between prothorax and mesothorax of insects. In the RNAi experiments, BPHs were then treated with the $LC_{50}$ of EB 24 h after dsRNA injection and the whole-body sampled for qRT-PCR analysis.

## Quantitative RT-PCR

Fourth-instar nymphs of BPH were treated with EB, after which total RNA was extracted from fifth-instar nymphs and 1–7 day post-eclosion females of *N. lugens* using the methods detailed above. The HiScript II RT SuperMix for qPCR (+gDNA wiper) kit from Vazyme, Nanjing, China, was used to create first-strand cDNA. Primer3 (http://bioinfo.ut.ee/primer3/) was used to design real-time quantitative PCR (qPCR) primers listed in *Supplementary file 2*. mRNA levels of candidate genes were detected by qPCR using the UltraSYBR Mixture (with ROX) Kit (CWBIO, Beijing, China). Each reaction contained 2 µl of cDNA template (500 ng), 1 µl each forward and reverse qPCR primer (10 µM), 10 µl of UltraSYBR mixture buffer, and 6 µl of RNase-free water. Q-PCR was run on an ABI 7500 Real-Time PCR System (Applied Biosystems, Foster City, CA, USA) under the following conditions: 5 min at 95°C, followed by 40 cycles of 95°C for 10 s and 60°C for 30 s. Three independent biological replicates and four technical replicates were used in each qPCR experiment. The housekeeping gene 18S ribosomal RNA was used to normalize the expression of candidate genes. The $2^{-\Delta\Delta Ct}$ method (Ct represents the cycle threshold) was used to measure relative expression levels (*Livak and Schmittgen, 2001*). Three biological replicates were used for statistical comparison between samples. *Supplementary file 2* contains a list of the primers used in this study.

## Gene expression quantification following RNAi

In late fourth-instar nymphs, we injected 40 nl of synthesized dsRNA targeting *GFP*, *JHAMT*, or *Kr-h1* (5000 ng/µl). After a 24 h recovery period, we applied EB using the systemic route described previously, while control rice seedlings were treated with a 0.1% Triton X-100 aqueous solution. 72 h post-treatment, we collected female BPHs from each treatment group (at least four groups, n≥5 per group) for total RNA extraction and first-strand cDNA synthesis. We then performed qPCR to measure mRNA levels of the candidate genes *JHAMT* and *Kr-h1* using the UltraSYBR Mixture (with ROX) kit (Beijing Zhongwei Biotech Co., China). The primers for dsRNA synthesis and the qPCR primers were identical to those used in the previous experiments.

## Statistics

PoloPlus v2.0 (LeOra Software 2008) was used to calculate the lethal concentration ($LC_{50}$) and 95% fiducial limits (95% F.L.) from bioassay data. GraphPad Prism 8.0 software (GraphPad Software Inc, San Diego, USA) was used to generate graphs and perform statistical analysis of data. The data presented in this study were first verified for normal distribution using the D'Agostino–Pearson normality test. One-way analysis of variance (ANOVA) with Duncan's multiple range test was used to test differences among multiple groups of normally distributed groups of data. Student's *t*-test was used to test the differences between two groups. If data was not normally distributed, Mann–Whitney tests were used for pairwise comparisons, and Kruskal–Wallis test was used for comparisons among multiple groups, followed by Dunn's multiple comparisons. All data are presented in figures as mean ± s.e.m. The sample sizes and statistical tests used for each experiment are stated in the figures or figure legends.

## Acknowledgements

This research was funded by the National Key R&D Program of China to SFW (2022YFD1700200) (https://service.most.gov.cn/) and National Natural Science Foundation of China to SFW (32472542) (https://www.nsfc.gov.cn/). The funders had no role in study design, data collection and analysis, decision to publish, or preparation of the manuscript.

# Additional information

## Funding

| Funder | Grant reference number | Author |
|---|---|---|
| National Key Research and Development Program of China | 2022YFD1700200 | Shun-Fan Wu |
| National Natural Science Foundation of China | 32472542 | Shun-Fan Wu |

The funders had no role in study design, data collection and interpretation, or the decision to submit the work for publication.

## Author contributions

Yang Gao, Data curation, Software, Formal analysis, Investigation, Methodology; Shao-Cong Su, Data curation, Software, Validation, Investigation, Methodology; Ji-Yang Xing, Resources, Data curation, Software, Validation, Investigation, Methodology; Zhao-Yu Liu, Data curation, Investigation, Methodology; Dick R Nässel, Chris Bass, Supervision, Writing – review and editing; Congfen Gao, Resources, Supervision, Validation; Shun-Fan Wu, Conceptualization, Resources, Data curation, Formal analysis, Supervision, Funding acquisition, Validation, Visualization, Writing – original draft, Project administration, Writing – review and editing

## Author ORCIDs

Shao-Cong Su ⓘ https://orcid.org/0009-0007-3782-6657
Dick R Nässel ⓘ https://orcid.org/0000-0002-1147-7766
Chris Bass ⓘ https://orcid.org/0000-0002-2590-1492
Shun-Fan Wu ⓘ https://orcid.org/0000-0003-0096-147X

Reviewer #1 (Public review): https://doi.org/10.7554/eLife.91774.4.sa1
Author response https://doi.org/10.7554/eLife.91774.4.sa2

---

# Additional files

## Supplementary files

Supplementary file 1. Determination of the toxicity of emamectin benzoate on BPH in systemic and topical application bioassays.

Supplementary file 2. Sequences of oligonucleotide primers used in this study.

Supplementary file 3. Amino acid sequences of neuropeptide G Protein-Coupled Receptors (GPCRs) and related receptors from multiple arthropod species.

Supplementary file 4. Amino acid sequences of Glutamate-gated Chloride channels (GluCls) and acetylcholine receptors from diverse invertebrate species.

MDAR checklist

## Data availability

All data generated or analysed during this study are included in the manuscript and supporting files; source data files have been provided for all figures.

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
