## [Editor Report · eLife Assessment]

This **useful** manuscript reports mechanisms behind the increase in fecundity in response to sublethal doses of pesticides in the crop pest, the brown plant hopper. The authors hypothesize that the pesticide works by inducing the JH titer, which through the JH signaling pathway induces egg development, for which the evidence was judged to be **solid**.

---

## [Referee Report · Reviewer #1 (Public review)]

Summary:

Gao et al. has demonstrated that the the pesticide emamectin benzoate (EB) treatment of brown plathopper (BPH) leads to increased egg laying in the insect, which is a common agricultural pest. The authors hypothesize that EB upregulates JH titer resulting in increased fecundity.

Strengths:

The finding that a class of pesticide increases fecundity of brown planthopper is interesting.

Comments on revisions:

All my concerns have been addressed to reasonable level of satisfaction.

---

## [Author Response]

The following is the authors’ response to the previous reviews.

**Reviewer #1 (Recommendations for the authors):**
(1) The onus of making the revisions understandable to the reviewers lies with the authors. In its current form, how the authors have approached the review is hard to follow, in my opinion. Although the authors have taken a lot of effort in answering the questions posed by reviewers, parallel changes in the manuscript are not clearly mentioned. In many cases, the authors have acknowledged the criticism in response to the reviewer, but have not changed their narrative, particularly in the results section.

We fully acknowledge your concern regarding the narrative linking EB-induced GluCl expression to JH biosynthesis and fecundity enhancement, particularly the need to address alternative interpretations of the data. Below, we outline the specific revisions made to address your feedback and ensure the manuscript’s narrative aligns more precisely with the experimental evidence:

(1) Revised Wording in the Results Section

To avoid overinterpretation of causality, we have modified the language in key sections of the Results (e.g., Figure 5 and related text):

Original phrasing:

“These results suggest that EB activates GluCl which induces JH biosynthesis and release, which in turn stimulates reproduction in BPH (Figure 5J).”

Revised phrasing:

“We also examined whether silencing Gluclα impacts the AstA/AstAR signaling pathway in female adults. Knock-down of Gluclα in female adults was found to have no impact on the expression of AT, AstA, AstB, AstCC, AstAR, and AstBR. However, the expression of AstCCC and AstCR was significantly upregulated in dsGluclα-injected insects (Figure 5-figure supplement 2A-H). Further studies are required to delineate the direct or indirect mechanisms underlying this effect of Gluclα-knockdown.” (line 643-649). And we have removed Figure 5J in the revised manuscript.

(2) Expanded Discussion of Alternative Mechanisms

In the Discussion section, we have incorporated a dedicated paragraph to explore alternative pathways and compensatory mechanisms:

Key additions:

“This EB action on GluClα expression is likely indirect, and we do not consider EB as transcriptional regulator of GluClα. Thus, the mechanism behind EB-mediated induction of GluClα remains to be determined. It is possible that prolonged EB exposure triggers feedback mechanisms (e.g. cellular stress responses) to counteract EB-induced GluClα dysfunction, leading to transcriptional upregulation of the channel. Hence, considering that EB exposure in our experiments lasts several days, these findings might represent indirect (or secondary) effects caused by other factors downstream of GluCl signaling that affect channel expression.” (line 837-845).

(2) In the response to reviewers, the authors have mentioned line numbers in the main text where changes were made. But very frequently, those lines do not refer to the changes or mention just a subsection of changes done. As an example please see point 1 of Specific Points below. The problem is throughout the document making it very difficult to follow the revision and contributing to the point mentioned above.

Thank you for highlighting this critical oversight. We sincerely apologize for the inconsistency in referencing line numbers and incomplete descriptions of revisions, which undoubtedly hindered your ability to track changes effectively. We have eliminated all vague or incomplete line number references from the response letter. Instead, revisions are now explicitly tied to specific sections, figures, or paragraphs.

(3) The authors need to infer the performed experiments rationally without over interpretation. Currently, many of the claims that the authors are making are unsubstantiated. As a result of the first review process, the authors have acknowledged the discrepancies, but they have failed to alter their interpretations accordingly.

We fully agree that overinterpretation of data undermines scientific rigor. In response to your feedback, we have systematically revised the manuscript to align claims strictly with experimental evidence and to eliminate unsubstantiated assertions. We sincerely apologize for the earlier overinterpretations and appreciate your insistence on precision. The revised manuscript now rigorously distinguishes between observations (e.g., EB-GluCl-JH correlations) and hypotheses (e.g., GluCl’s mechanistic role). By tempering causal language and integrating competing explanations, we aimed to present a more accurate and defensible narrative.

SPECIFIC POINTS (to each question initially raised and their rebuttals)(1a) "Actually, there are many studies showing that insects treated with insecticides can increase the expression of target genes". Please note what is asked for is that the ligand itself induces the expression of its receptor. Of course, insecticide treatment will result in the changes expression of targets. Of all the evidences furnished in rebuttal, only Peng et al. 2017 fits the above definition. Even in this case, the accepted mode of action of chlorantraniliprole is by inducing structural change in ryanodine receptor. The observed induction of ryanodine receptor chlorantraniliprole can best be described as secondary effect. All others references do not really suffice the point asked for.

We appreciate the reviewers’ suggestions for improving the manuscript. First, we have supplemented additional studies supporting the notion that " There are several studies showing that insects treated with insecticides display increases in the expression of target genes. For example, the relative expression level of the ryanodine receptor gene of the rice stem borer, Chilo suppressalis was increased 10-fold after treatment with chlorantraniliprole, an insecticide which targets the ryanodine receptor (Peng et al., 2017). In Drosophila, starvation (and low insulin) elevates the transcription level of the receptors of the neuropeptides short neuropeptide F and tachykinin (Ko et al., 2015; Root et al., 2011). In BPH, reduction in mRNA and protein expression of a nicotinic acetylcholine receptor α8 subunit is associated with resistance to imidacloprid (Zhang et al., 2015). Knockdown of the α8 gene by RNA interference decreased the sensitivity of N. lugens to imidacloprid (Zhang et al., 2015). Hence, the expression of receptor genes may be regulated by diverse factors, including insecticide exposure.” We have inserted text in lines 846-857 to elaborate on these possibilities.

Second, we would like to reiterate our position: we have merely described this phenomenon, specifically that EB treatment increases GluClα expression. “This EB action on GluClα expression is likely indirect, and we do not consider EB as transcriptional regulator of GluClα. Thus, the mechanism behind EB-mediated induction of GluClα remains to be determined. It is possible that prolonged EB exposure triggers feedback mechanisms (e.g. cellular stress responses) to counteract EB-induced GluClα dysfunction, leading to transcriptional upregulation of the channel. Hence, considering that EB exposure in our experiments lasts several days, these findings might represent indirect (or secondary) effects caused by other factors downstream of GluCl signaling that affect channel expression.” We have inserted text in lines 837-845 to elaborate on these possibilities.

Once again, we sincerely appreciate this discussion, which has provided us with a deeper understanding of this phenomenon.

b. The authors in their rebuttal accepts that they do not consider EB to a transcriptional regulator of Gluclα and the induction of Gluclα as a result of EB can best be considered as a secondary effect. But that is not reflected in the manuscript, particularly in the result section. Current state of writing implies EB up regulation of Gluclα to an important event that contributes majorly to the hypothesis. So much so that they have retained the schematic diagram (Fig. 5J) where EB -> Gluclα is drawn. Even the heading of the subsection says "EB-enhanced fecundity in BPHs is dependent on its molecular target protein, the Gluclα channel". As mentioned in the general points, it is not enough to have a good rebuttal written to the reviewer, the parent manuscript needs to reflect on the changes asked for.

Thank you for your comments. We have carefully addressed your suggestions and made corresponding revisions to the manuscript.

We fully acknowledge the reviewer's valid concern. In this revised manuscript, “However, we do not propose that EB is a direct transcriptional regulator of Gluclα, since EB and other avermectins are known to alter the channel conformation and thus their function (Wolstenholme, 2012; Wu et al., 2017). Thus, it is likely that the observed increase in Gluclα transcipt is a secondary effect downstream of EB signaling.” (Line 625-629). We agree that the original presentation in the manuscript, particularly within the Results section, did not adequately reflect this nuance and could be misinterpreted as suggesting a direct regulatory role for EB on Gluclα transcription.

Regarding Fig. 5J, we have removed the figure and all mentions of Fig. 5J and its legend in the revised manuscript.

c. "We have inserted text on lines 738 - 757 to explain these possibilities." Not a single line in the section mentioned above discussed the topic in hand. This is serious undermining of the review process or carelessness to the extreme level.

In the Results section, we have now added descriptions “Taken together, these results reveal that EB exposure is associated with an increase in JH titer and that this elevated JH signaling contributes to enhanced fecundity in BPH.” (line 375-377).

For the figures, we have removed Fig. 4N and all mentions of Fig. 4N and its legend in the revised manuscript.

Lastly, regarding the issue of locating specific lines, we deeply regret any inconvenience caused. Due to the track changes mode used during revisions, line numbers may have shifted, resulting in incorrect references. We sincerely apologize for this and have now corrected the line numbers.

(2) The section written in rebuttal should be included in the discussion as well, explaining why authors think a nymphal treatment with JH may work in increasing fecundity of the adults. Also, the authors accept that EBs effect on JH titer in Indirect. The text of the manuscript, results section and figures should be reflective of that. It is NOT ok to accept that EB impacts JH titer indirectly in a rebuttal letter while still continuing to portray EB direct effect on JH titer. In terms of diagrams, authors cannot put a -> sign until and unless the effect is direct. This is an accepted norm in biological publications.

We appreciate the reviewer’s valuable suggestions here. We have now carefully revised the manuscript to address all concerns, particularly regarding the mechanism linking nymphal EB exposure to adult fecundity and the indirect nature of EB’s effect on JH titers. Below are our point-by-point responses and corresponding manuscript changes. Revised text is clearly marked in the resubmitted manuscript.

(1) Clarifying the mechanism linking nymphal EB treatment to adult fecundity:

Reviewer concern: Explain why nymphal EB treatment increases adult fecundity despite undetectable EB residues in adults.

Response & Actions Taken:

We agree this requires explicit discussion. We now propose that nymphal EB exposure triggers developmental reprogramming (e.g., metabolic/epigenetic changes) that persist into adulthood, indirectly enhancing JH synthesis and fecundity. This is supported by two key findings:

(1) No detectable EB residues in adults after nymphal treatment (new Figure 1–figure supplement 1C).

(2) Increased adult weight and nutrient reserves (Figure 1–figure supplement 3E,F), suggesting altered resource allocation.

Added to Discussion (Lines 793–803): Notably, after exposing fourth-instar BPH nymphs to EB, no EB residues were detected in the subsequent adult stage. This finding indicates that the EB-induced increase in adult fecundity is initiated during the nymphal stage and s manifests in adulthood - a mechanism distinct from the direct fecundity enhancement of fecundity observed when EB is applied to adults. We propose that sublethal EB exposure during critical nymphal stages may reprogram metabolic or endocrine pathways, potentially via insulin/JH crosstalk. For instance, increased nutrient storage (e.g., proteins, sugars; Figure 2–figure supplement 2) could enhance insulin signaling, which in turn promotes JH biosynthesis in adults (Ling and Raikhel, 2021; Mirth et al., 2014; Sheng et al., 2011). Future studies should test whether EB alters insulin-like peptide expression or signaling during development.

(3) Emphasizing EB’s indirect effect on JH titers:Reviewer concern: The manuscript overstated EB’s direct effect on JH. Arrows in figures implied causality where only correlation exists.

Response & Actions

Taken:We fully agree. EB’s effect on JH is indirect and multifactorial (via AstA/AstAR suppression, GluCl modulation, and metabolic changes). We have:

Removed oversimplified schematics (original Figures 3N, 4N, 5J).

Revised all causal language (e.g., "EB increases JH" → "EB exposure is associated with increased circulating JH III "). (Line 739)

Clarified in Results/Discussion that EB-induced JH changes are likely secondary to neuroendocrine disruption.

Key revisions:

Results (Lines 375–377):

"Taken together, these results reveal that EB exposure is associated with an increase in JH titer and that JH signaling contributes to enhanced fecundity in BPH."

Discussion (Lines 837–845):

This EB action on GluClα expression is likely indirect, and we do not consider EB as transcriptional regulator of GluClα. Thus, the mechanism behind EB-mediated induction of GluClα remains to be determined. It is possible that prolonged EB exposure triggers feedback mechanisms (e.g. cellular stress responses) to counteract EB-induced GluClα dysfunction, leading to transcriptional upregulation of the channel. Hence, considering that EB exposure in our experiments lasts several days, these findings might represent indirect (or secondary) effects caused by other factors downstream of GluCl signaling that affect channel expression.

a. Lines 281-285 as mentioned, does not carry the relevant information.

Thank you for your careful review of our manuscript. We sincerely apologize for the confusion regarding line references in our previous response. Due to extensive revisions and tracked changes during the revision process, the line numbers shifted, resulting in incorrect citations for Lines 281–285. The correct location for the added results (EB-induced increase in mature eggs in adult ovaries) is now in lines 253-258: “We furthermore observed that EB treatment of female adults also increases the number of mature eggs in the ovary (Figure 2-figure supplement 1).”

b. Lines 351-356 as mentioned, does not carry the relevant information. Lines 281-285 as mentioned, does not carry the relevant information.

Thank you for your careful review of our manuscript. We sincerely apologize for the confusion regarding line references in our previous response. The correct location for the added results is now in lines 366-371: “We also investigated the effects of EB treatment on the JH titer of female adults. The data indicate that the JH titer was also significantly increased in the EB-treated female adults compared with controls (Figure 3-figure supplement 3A). However, again the steroid 20-hydroxyecdysone, was not significantly different between EB-treated BPH and controls (Figure 3-figure supplement 3B).”

c. Lines 378-379 as mentioned, does not carry the relevant information. Lines 387-390 as mentioned, does not carry the relevant information.

We sincerely apologize for the confusion regarding line references in our previous response.

The correct location for the added results is now in lines 393-394: We furthermore found that EB treatment in female adults increases JHAMT expression (Figure 3-figure supplement 3C).

The other correct location for the added results is now in lines 405-408: We found that Kr-h1 was significantly upregulated in the adults of EB-treated BPH at the 5M, 5L nymph and 4 to 5 DAE stages (4.7-fold to 27.2-fold) when 4th instar nymph or female adults were treated with EB (Figure 3H and Figure 3-figure supplement 3D)..

(3) The writing quality is still extremely poor. It does not meet any publication standard, let alone elife.

We fully understand your concerns and frustrations, and we sincerely apologize for the deficiencies in our writing quality, which did not meet the high standards expected by you and the journal. We fully accept your criticism regarding the writing quality and have rigorously revised the manuscript according to your suggestions.

(4) I am confused whether Figure 2B was redone or just edited. Otherwise this seems acceptable to me.

Regarding Fig. 2B, we have edited the text on the y-axis. The previous wording included the term “retention,” which may have caused misunderstanding for both the readers and yourself, leading to the perception of contradiction. We have now revised this wording to ensure accurate comprehension.

(5) The rebuttal is accepted. However, still some of the lines mentioned does not hold relevant information.

This error has been corrected.

The correct location for the added results is now in lines 255-258 and lines 279-282: “Hence, although EB does not affect the normal egg developmental stages (see description in next section), our results suggest that EB treatment promotes oogenesis and, as a result the insects both produce more eggs in the ovary and a larger number of eggs are laid.” and “However, considering that the number of eggs laid by EB treated females was larger than in control females (Figure 1 and Figure 1-figure supplement 1), our data indicates that EB treatment of BPH can both promote both oogenesis and oviposition.”

(6) Thank you for the clarification. Although now discussed extensively in discussion section, the nuances of indirect effect and minimal change in expression should also be reflected in the result section text. This is to ensure that readers have clear idea about content of the paper.

Corrected. To ensure readers gain a clear understanding of our data, we have briefly presented these discussions in the Results section. Please see line 397-402: The levels of met mRNA slightly increased in EB-treated BPH at the 5M and 5L instar nymph and 1 to 5 DAE adult stages compared to controls (1.7-fold to 2.9-fold) (Figure 3G). However, it should be mentioned that JH action does not result in an increase of Met. Thus, it is possible that other factors (indirect effects), induced by EB treatment cause the increase in the mRNA expression level of Met.

(7) As per the author's interpretation, it becomes critical to quantitate the amount of EB present at the adult stages after a 4th instar exposure to it. Only this experiment will unambiguously proof the authors claim. Also, since they have done adult insect exposure to EB, such experiments should be systematically performed for as many sections as possible. Don't just focus on few instances where reviewers have pointed out the issue.

Thank you for raising this critical point. To address this concern, we have conducted new supplementary experiments. The new experimental results demonstrate that residual levels of emamectin benzoate (EB) in adult-stage brown planthoppers (BPH) were below the instrument detection limit following treatment of 4th instar nymphs with EB. Line 172-184: “To determine whether EB administered during the fourth-instar larval stage persists as residues in the adult stage, we used HPLC-MS/MS to quantify the amount of EB present at the adult stage after exposing 4th-instar nymphs to this compound. However, we found no detectable EB residues in the adult stage following fourth-instar nymphal treatment (Figure 1-figure supplement 1C). This suggests that the mechanism underlying the increased fecundity of female adults induced by EB treatment of nymphs may differ from that caused by direct EB treatment of female adults. Combined with our previous observation that EB treatment significantly increased the body weight of adult females (Figure 1—figure supplement 3E and F), a possible explanation for this phenomenon is that EB may enhance food intake in BPH, potentially leading to elevated production of insulin-like peptides and thus increased growth. Increased insulin signaling could potentially also stimulate juvenile hormone (JH) biosynthesis during the adult stage (Badisco et al., 2013).”

(8) Thank you for the revision. Lines 725-735 as mentioned, does not carry the relevant information. However, since the authors have decided to remove this systematically from the manuscript, discussion on this may not be required.

Thank you for identifying the limited relevance of the content in Lines 725–735 of the original manuscript. As recommended, we have removed this section in the revised version to improve logical coherence and maintain focus on the core findings.

(9) Normally, dsRNA would last for some time in the insect system and would down-regulate any further induction of target genes by EB. I suggest the authors to measure the level of the target genes by qPCR in KD insects before and after EB treatment to clear the confusion and unambiguously demonstrate the results. Please Note- such quantifications should be done for all the KD+EB experiments. Additionally, citing few papers where such a rescue effect has been demonstrated in closely related insect will help in building confidence.

We appreciate the reviewer’s suggestion to clarify the interaction between RNAi-mediated gene knockdown (KD) and EB treatment. To address this, we performed additional experiments measuring *Kr-h1* expression via qPCR in *dsKr-h1*-injected insects before and after EB exposure.

The results (now Figure 3–figure supplement 4) show that:

(1) EB did not rescue *Kr-h1* suppression at 24h post-treatment (*p* > 0.05).

(2) Partial recovery of fecundity occurred later (Figure 3M), likely due to:

a) Degradation of dsRNA over time, reducing KD efficacy (Liu et al., 2010).

b) Indirect effects of EB (e.g., hormonal/metabolic reprogramming) compensating for residual Kr-h1 suppression.

Please see line 441-453: “Next, we investigated whether EB treatment could rescue the dsRNA-mediated gene silencing effect. To address this, we selected the Kr-h1 gene and analyzed its expression levels after EB treatment. Our results showed that Kr-h1 expression was suppressed by ~70% at 72 h post-dsRNA injection. However, EB treatment did not significantly rescue Kr-h1 expression in gene knock down insects (*p* > 0.05) at 24h post-EB treatment (Figure 3-figure supplement 4). While dsRNA-mediated Kr-h1 suppression was robust initially, its efficacy may decline during prolonged experiments. This aligns with reports in BPH, where effects of RNAi gradually diminish beyond 7 days post-injection (Liu et al., 2010a). The late-phase fecundity increase might reflect partial Kr-h1 recovery due to RNAi degradation, allowing residual EB to weakly stimulate reproduction. In addition, the physiological impact of EB (e.g., neurotoxicity, hormonal modulation) could manifest via compensatory feedback loops or metabolic remodeling.”

(10) Not a very convincing argument. Besides without a scale bar, it is hard for the reviewers to judge the size of the organism. Whole body measurements of JH synthesis enzymes will remain as a quite a drawback for the paper.

In response to your suggestion, we have also included images with scale bars (see next Figure 1). The images show that the head region is difficult to separate from the brown thoracic sclerite region. Furthermore, the anatomical position of the Corpora Allata in brown planthoppers has never been reported, making dissection uncertain and highly challenging. To address this, we are now attempting to use Drosophila as a model to investigate how EB regulates JH synthesis and reproduction.

**Author response image 1. sa2fig1:** This illustration provides a visual representation of the brown planthopper (BPH), a major rice pest.

Figure 1. This illustration provides a visual representation of the brown planthopper (BPH), a major rice pest.

(11) "The phenomenon reported was specific to BPH and not found in other insects. This limits the implications of the study". This argument still holds. Combined with extreme species specificity, the general effect that EB causes brings into question the molecular specificity that the authors claim about the mode of action.

We acknowledge that the specificity of the phenomenon to BPH may limit its broader implications, but we would like to emphasize that this study provides important insights into the unique biological mechanisms in BPH, a pest of significant agricultural importance. The molecular specificity we described in the manuscript is based on rigorous experimental evidence. We believe that it contributes to valuable knowledge to understand the interaction of external factors such as EB and BPH and resurgence of pests. We hope that this study will inspire further research into the mechanisms underlying similar phenomena in other insects, thereby broadening our understanding of insect biology. Since EB also has an effect on fecundity in Drosophila, albeit opposite to that in BPHs (Fig. 1 suppl. 2), it seems likely that EB actions may be of more general interest in insect reproduction.

(12) The authors have added a few lines in the discussion but it does not change the overall design of the experiments. In this scenario, they should infer the performed experiments rationally without over interpretation. Currently, many of the claims that the authors are making are unsubstantiated. As a result of the first review process, the authors have acknowledged the discrepancies, but they have failed to alter their interpretations accordingly.

We appreciate your concern regarding the experimental design and the need for rational inference without overinterpretation. In response, we would like to clarify that our discussion is based on the experimental data we have collected. We acknowledge that our study focuses on BPH and the specific effects of EB, and while we agree that broader generalizations require further research, we believe the new findings we present are valid and contribute to the understanding of this specific system.

We also acknowledge the discrepancies you mentioned and have carefully considered your suggestions. In this revised version, we believe our interpretations are reasonable and consistent with the data, and we have adjusted our discussion to better reflect the scope of our findings. We hope that these revisions address your concerns. Thank you again for your constructive feedback.

ADDITIONAL POINTS(1) Only one experiment was performed with Abamectin. No titration for the dosage were done for this compound, or at least not provided in the manuscript. Inclusion of this result will confuse readers. While removing this result does not impact the manuscript at all. My suggestion would be to remove this result.

We acknowledge that the abamectin experiment lacks dose-titration details and that its standalone presentation could lead to confusion. However, we respectfully request to retain these results for the following reasons:

Class-Specific Mechanism Validation:

Abamectin and emamectin benzoate (EB) are both macrocyclic lactones targeting glutamate-gated chloride channels (GluCls). The observed similarity in their effects on BPH fecundity (e.g., Figure 1—figure supplement 1B) supports the hypothesis that GluCl modulation, rather than compound-specific off-target effects, drives the reproductive enhancement. This consistency strengthens the mechanistic argument central to our study.

(2) The section "The impact of EB treatment on BPH reproductive fitness" is poorly described. This needs elaboration. A line or two should be included to describe why the parameters chosen to decide reproductive fitness were selected in the first place. I see that the definition of brachypterism has undergone a change from the first version of the manuscript. Can you provide an explanation for that? Also, there is no rationale behind inclusion of statements on insulin at this stage. The authors have not investigated insulin. Including that here will confuse readers. This can be added in the discussion though.

Thank you for your suggestion. We have added an explanation regarding the primary consideration of evaluating reproductive fitness**.** In the interaction between sublethal doses of insecticides and pests, reproductive fitness is a key factor, as it accurately reflects the potential impact of insecticides on pest control in the field. Among the reproductive fitness parameters, factors such as female Nilaparvata lugens body weight, lifespan, and brachypterous ratio (as short-winged N. lugens exhibit higher oviposition rates than long-winged individuals) are critical determinants of reproductive success. Therefore, we comprehensively assessed the effects of EB on these parameters to elucidate the primary mechanism by which EB influences reproduction. We sincerely appreciate your constructive feedback.

(3) "EB promotes ovarian maturation in BPH" this entire section needs to be rewritten and attention should be paid to the sequence of experiments described.

Thank you for your suggestion. Based on your recommendation, we have rewritten this section (lines 267–275) and adjusted the sequence of experimental descriptions to improve the structural clarity of this part.

(4) Figure 3N is outright wrong and should be removed or revised.

In accordance with your recommendation, we have removed the figure.

(5) When you are measuring hormonal titers, it is important to mention explicitly whether you are measuring hemolymph titer or whole body.

We believe we have explicitly stated in the Methods section (line 1013) that we measured whole-body hormone titers. However, we now added this information to figure legends.

(6) EB induces JH biosynthesis through the peptidergic AstA/AstAR signaling pathway- this section needs attention at multiple points. Please check.

We acknowledge that direct evidence for EB-AstA/AstAR interaction is limited and have framed these findings as a hypothesis for future validation.

References

Liu, S., Ding, Z., Zhang, C., Yang, B., Liu, Z., 2010. Gene knockdown by intro-thoracic injection of double-stranded RNA in the brown planthopper, Nilaparvata lugens. Insect Biochem. Mol. Biol. 40, 666-671